# Retinal metabolism displays evidence for uncoupling of glycolysis and oxidative phosphorylation via Cori-, Cahill-, and mini-Krebs-cycle

Yiyi Chen[1†], Laimdota Zizmare[2,3†], Victor Calbiague[4], Lan Wang[1], Shirley Yu[1], Fritz W Herberg[5], Oliver Schmachtenberg[4], Francois Paquet-Durand[1*‡], Christoph Trautwein[2,3*]

[1]Institute for Ophthalmic Research, University of Tübingen, Tuebingen, Germany; [2]Werner Siemens Imaging Center, Department of Preclinical Imaging and Radiopharmacy, University of Tübingen, Tuebingen, Germany; [3]Core Facility Metabolomics, Faculty of Medicine, University of Tübingen, Tuebingen, Germany; [4]Centro Interdisciplinario de Neurociencia de Valparaíso, Universidad de Valparaíso, Valparaíso, Chile; [5]Biochemistry Department, University of Kassel, Tuebingen, Germany

**\*For correspondence:**
francois.paquet-durand@klinikum.uni-tuebingen.de (FP-D);
christoph.trautwein@med.uni-tuebingen.de (CT)

[†]These authors contributed equally to this work

[‡]Lead contact

**Competing interest:** The authors declare that no competing interests exist.

**Abstract** The retina consumes massive amounts of energy, yet its metabolism and substrate exploitation remain poorly understood. Here, we used a murine explant model to manipulate retinal energy metabolism under entirely controlled conditions and utilised [1]H-NMR spectroscopy-based metabolomics, in situ enzyme detection, and cell viability readouts to uncover the pathways of retinal energy production. Our experimental manipulations resulted in varying degrees of photoreceptor degeneration, while the inner retina and retinal pigment epithelium were essentially unaffected. This selective vulnerability of photoreceptors suggested very specific adaptations in their energy metabolism. Rod photoreceptors were found to rely strongly on oxidative phosphorylation, but only mildly on glycolysis. Conversely, cone photoreceptors were dependent on glycolysis but insensitive to electron transport chain decoupling. Importantly, photoreceptors appeared to uncouple glycolytic and Krebs-cycle metabolism via three different pathways: (1) the mini-Krebs-cycle, fuelled by glutamine and branched chain amino acids, generating *N*-acetylaspartate; (2) the alanine-generating Cahill-cycle; (3) the lactate-releasing Cori-cycle. Moreover, the metabolomics data indicated a shuttling of taurine and hypotaurine between the retinal pigment epithelium and photoreceptors, likely resulting in an additional net transfer of reducing power to photoreceptors. These findings expand our understanding of retinal physiology and pathology and shed new light on neuronal energy homeostasis and the pathogenesis of neurodegenerative diseases.

## eLife assessment

Chen and colleagues utilize an in situ explant model of the neural retina and retinal pigment epithelium (RPE), along with small molecule inhibition of key metabolic enzymes and targeted metabolomic analysis, to decipher key differences in metabolic pathways used by rods, cones, Muller glia, and the RPE. They conclude that rods are heavily reliant on oxidative metabolism, cones are heavily reliant on glycolysis, and multiple mechanisms exist to decouple glycolysis from oxidative metabolism in the retina. This study provides **valuable** metabolomic data and insights into the metabolic flexibility of different retinal cells. However, current evidence is still **incomplete** as several of the

conclusions from the paper stand in contradiction to other published findings and the authors naturally suggests experiments that will be needed in the future to validate the hypothesized pathways and refute existing published data. Such future validation includes animal models with tissue specific knockout of the key enzymes probed in the study; inhibiting the targets of this study with more than 1 small molecule that is structurally different, and at different doses and timings; using retinal explants from matured animals; performing labeled metabolite tracing experiments; and direct assessment of mitochondrial function (via OCR) under various manipulations.

## Introduction

The retina is the neuronal tissue with the highest energy demand (*Country, 2017*; *Wong-Riley, 2010*). Paradoxically, the mammalian retina is thought to strongly rely on energy-inefficient glycolysis, even though oxygen is available and high-yield oxidative phosphorylation (OXPHOS) possible (*Swarup et al., 2019*). This aerobic glycolysis releases large amounts of lactate, as already reported in the 1920s by *Warburg, 1925*. High retinal energy demand is linked to the extraordinary single-photon sensitivity of photoreceptors (*Wong-Riley, 2010*; *Okawa et al., 2008*; *Ames, 1992*), and to lipid synthesis for the constant renewal of photoreceptor outer segments (*Chinchore et al., 2017*; *Young, 1967*).

The retina harbours two types of photoreceptors: rods, which exhibit remarkable light sensitivity and enable night vision; and cones, which work in daylight and allow colour vision. Cones spend around twice as much energy as rods (*Ingram et al., 2020*). Photoreceptors are not connected to the vasculature and are nourished by other retinal cell types, including retinal pigment epithelial (RPE) (*Strauss, 2005*) cells or Müller glial cells (MGCs). Moreover, photoreceptors can experience changes in energy demand on a millisecond timescale, from very high in the dark, to 4–10 times lower in light (*Okawa et al., 2008*).

Glycolysis provides for rapid but inefficient adenosine triphosphate (ATP) production, while Krebs-cycle and OXPHOS are very efficient but much slower. Since glycolysis and Krebs-cycle are metabolically coupled through pyruvate, it is unclear how photoreceptors adapt to sudden and large changes in energy demand, and what energy substrates they may use. A recent hypothesis proposed that photoreceptors use predominantly aerobic glycolysis, with the resultant lactate utilised by the RPE and MGCs for OXPHOS (*Kanow et al., 2017*). Such an arrangement would be reminiscent of the Cori-cycle, in which glucose and lactate are cycled between skeletal muscle and liver (*Cori and Cori, 1946*). However, this concept contrasts with the high density of mitochondria in photoreceptor inner segments, which strongly suggest an extensive use of OXPHOS.

Alternative pathways for ATP generation include the Cahill-cycle, where pyruvate is transaminated using, for instance, glutamate to yield alanine and α-ketoglutarate (*Cahill and Owen, 1968*; *Felig, 1973*). The advantage of Cahill- over Cori-cycle is that reducing equivalents (i.e. NADH) gained during glycolysis are preserved and that α-ketoglutarate can yield additional ATP in mitochondrial OXPHOS. A second alternative may be the so-called mini-Krebs-cycle, in which a transamination of oxaloacetate with glutamate produces aspartate and α-ketoglutarate (*Yudkoff et al., 1994*). This shortens the 'full' 10-step Krebs-cycle to a significantly faster 6-step cycle. A point that Cori-, Cahill-, and mini-Krebs-cycle have in common is that they do not require the pyruvate-coupling between cytoplasmic glycolysis and mitochondrial OXPHOS, allowing both processes to run independent of each other. Whatever the case in the retina, the question as to which pathway is used has obvious ramifications for disease pathogenesis, including for diabetic retinopathy, age-related macular degeneration, or inherited retinal diseases, such as retinitis pigmentosa.

Here, we studied the expression of energy metabolism-related enzymes in different retinal cell types, using organotypic retinal explants maintained in serum-free, fully defined medium. Metabolic functions of the RPE were investigated by culturing retina with and without RPE. As readouts, we correlated enzyme expression with quantitative proton nuclear magnetic resonance ($^1$H-NMR) spectroscopy-based metabolomics and cell death. Notably, interventions in energy metabolism caused selective photoreceptor cell death, while leaving other retinal cell types essentially unaffected. Metabolomic analysis and localisation of key enzymes identified important pathways and shuttles altered, explaining the strong interdependence of the various retinal cell types, and opening new perspectives for the treatment of neurodegenerative diseases in the retina and beyond.

## Results

### Retinal expression patterns of key energy metabolism-related enzymes

We used immunofluorescence to assess expression and cellular localisation of enzymes important for energy metabolism in wild-type (WT) mouse retina in vivo (*Figure 1*). The outermost retinal layer is formed by RPE cells expressing RPE-specific 65 kDa protein (RPE65), dedicated to the recycling of photopigment, retinal (*Redmond et al., 1998*). Hence, RPE65 immunolabelling revealed the RPE cell monolayer (*Figure 1A*). Immunofluorescence for glucose transporter-1 (GLUT1) showed strong labelling on basal and apical sides of RPE cells (*Figure 1A*), in line with previous literature (*Swarup et al., 2019*; *Takata et al., 1992*). GLUT1 expression was not detected in rods or cones (*Figure 1—figure supplement 1*). Instead, glucose uptake in the outer retina seems to be mediated by high-affinity/high-capacity GLUT3 (*Simpson et al., 2008*), strongly expressed on photoreceptor inner segments (*Figure 1—figure supplement 2A*). Pyruvate kinase is essential to glycolysis, catalyzing the conversion of phosphoenolpyruvate to pyruvate. While pyruvate kinase M1 (PKM1) was expressed in the inner retina (*Figure 1—figure supplement 2A*), PKM2 was found in photoreceptor inner segments and synapses (*Figure 1A*). Expression of mitochondrial cytochrome oxidase (COX) largely overlapped with PKM2 (*Figure 1A*).

### Challenging energy metabolism reduces rod and cone photoreceptor viability

To dissect retinal energy metabolism, we selectively manipulated key pathways (*Figure 1B*) using organotypic retinal explants (*Belhadj et al., 2020*; *Figure 1—figure supplement 2B*). Retinal cultures were prepared with the RPE cell layer attached to the neuroretina (control) or without RPE (*no* RPE). The in vivo expression patterns of enzymes important for energy metabolism (*Figure 1A*) were comparable in the in vitro situation (*Figure 1—figure supplement 2C*). In control, the rate of cell death in the outer nuclear layer (ONL), as assessed by the terminal dUTP-nick-end labelling (TUNEL) assay (*Figure 1C and D*), was low (0.46%±0.22, n=9). The *no* RPE condition displayed significantly increased photoreceptor cell death (3.35%±1.62, n=5, p<0.001). Blocking RPE glucose transport with the selective GLUT1 inhibitor 1,9-dideoxyforskolin (1,9-DDF) (*Simpson et al., 2008*) further increased ONL cell death (6.77%±1.57, n=5, p<0.0001). The inhibition of glycolytic PKM2 with Shikonin (*Zhao et al., 2018*) and OXPHOS disruption with the electron chain uncoupler carbonyl cyanide-*p*-trifluoromethoxyphenylhydrazone (FCCP) (*Redmond et al., 1998*) both caused a significant increase in photoreceptor cell death (Shikonin: 17.75%±2.13, n=5, p<0.0001; FCCP: 22.49%±6.79, n=5, p<0.0001). All four experimental interventions mostly affected photoreceptors, not significantly reducing the viability of other retinal cell types (*Figure 1C*).

Remarkably, a differential effect on cone photoreceptor survival (*Figure 1E and F*) was observed as assessed by immunodetection of the cone-specific marker arrestin-3 (Arr3). Control retina harboured 11.47 (±1.95, n=9) cones per 100 µm of retinal circumference. *no* RPE retina displayed a significant decrease in cone survival (4.62±1.39, n=5, p<0.0001), while blocking RPE glucose transport with 1,9-DDF led to an even more dramatic reduction (1.02±1.09, n=5, p<0.0001). By comparison, inhibition of glycolysis with Shikonin had a relatively mild effect on cone survival (7.78±1.36, n=5, p<0.0001). Surprisingly, FCCP treatment did not cause cone death (12.84±1.75, n=5, p=0.22), compared to control.

We further calculated the percentages of dying cones and rods, assuming that 3% of all photoreceptors were cones (*Ortín-Martínez et al., 2014*; *Jeon et al., 1998*; *Figure 1G and H*). In the four treatment situations the ratios of cone to rod death were: *no* RPE = 1:1; 1,9-DDF = 1:1.5; Shikonin = 1:17; and FCCP = 0: ∞. Cones were almost entirely depleted by the 1,9-DDF treatment but remained virtually unaffected by the FCCP treatment. Rods, however, were strongly and highly selectively affected by Shikonin and FCCP, while the *no* RPE condition had a relatively minor effect. These results highlight important differences between rod and cone energy metabolism.

### Experimental retinal interventions produce characteristic metabolomic patterns

We employed high-field (600 MHz) $^1$H-NMR spectroscopy-based metabolomics to study the metabolic signatures in the five experimental groups (*Table 1Figure 2—figure supplement 1*). A principal

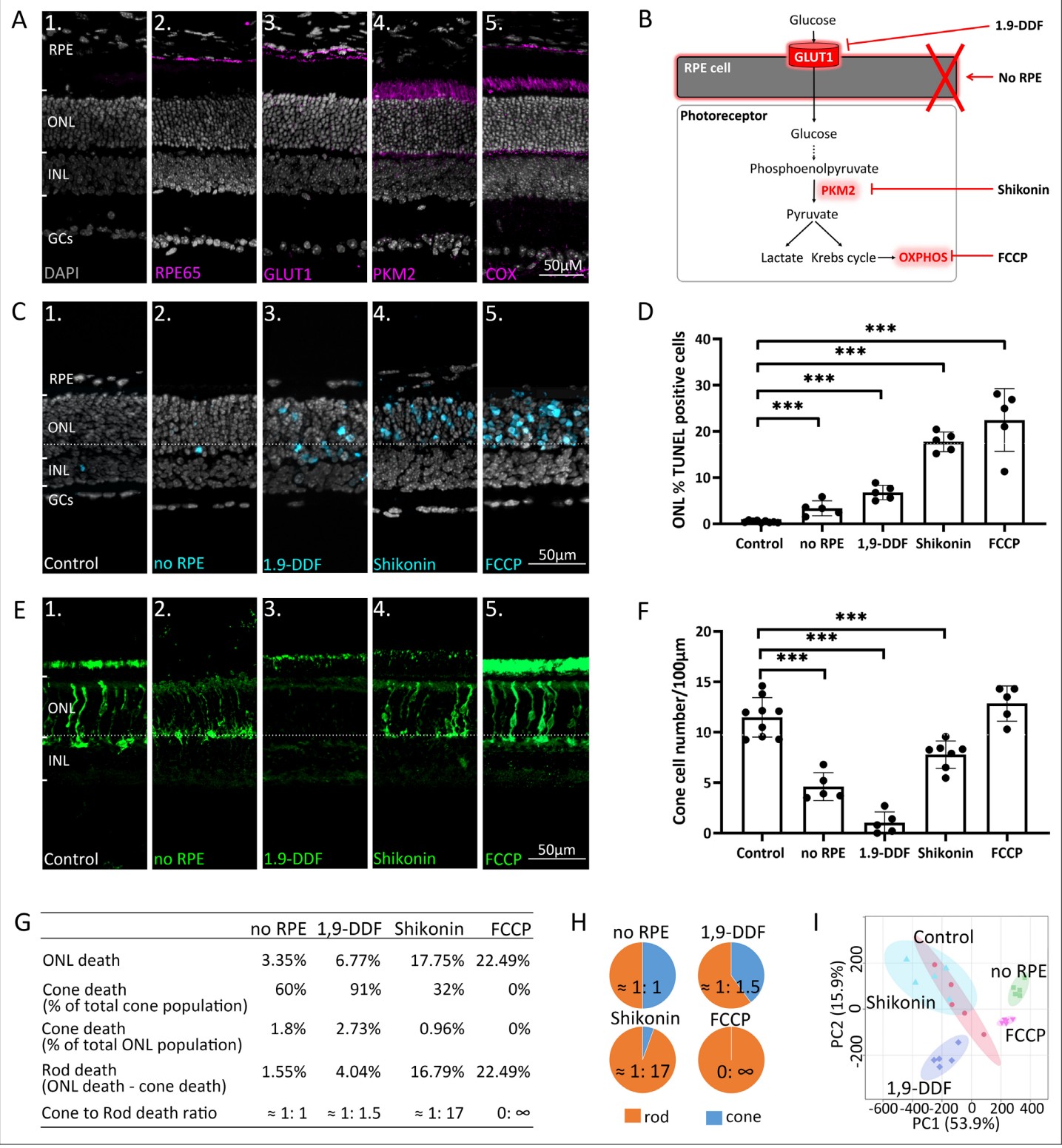

**Figure 1.** Manipulating energy metabolism differentially affects rod and cone photoreceptors. (**A**) Immunofluorescence staining of in vivo retina (magenta): (**A1**) negative control, (**A2**) RPE65, (**A3**) glucose transporter-1 (GLUT1), (**A4**) pyruvate kinase M2 (PKM2), (**A5**) mitochondrial cytochrome oxidase (COX). (**B**) Overview of experimental manipulations. Organotypic retinal explants were cultured for 6 days in vitro with or without RPE or treated with 1,9-dideoxyforskolin (1,9-DDF), Shikonin, or carbonyl cyanide-*p*-trifluoromethoxyphenylhydrazone (FCCP). (**C**) Terminal dUTP-nick-end labelling (TUNEL) assay performed on in vitro retina (cyan) marking dying cells in the five experimental conditions. (**D**) Quantification of TUNEL-positive cells in outer nuclear layer (ONL). Data represented as mean ± SD. (**E**) Cone-arrestin labelling (green) in the ONL. (**F**) Quantification of arrestin-positive cones.

*Figure 1 continued on next page*

*Figure 1 continued*

Data represented as mean ± SD. (**G**) Table giving percentages of cones and rods lost for each treatment. (**H**) Pie charts illustrate cone to rod cell death ratios. (**I**) Principal component analysis (PCA) of retinal samples investigated with $^1$H-NMR spectroscopy-based metabolomics. Dots in graphs represent individual retinal explants from different animals (n=5). Statistical testing: one-way ANOVA with Tukey's post hoc test. Asterisks indicate significance levels; p-values: ***<0.001, **<0.01, *<0.05. RPE = retinal pigment epithelium; INL = inner nuclear layer; GCs = ganglion cells.

The online version of this article includes the following figure supplement(s) for figure 1:

**Figure supplement 1.** Co-staining of glucose transporter-1 (GLUT1) with markers for rod and cone photoreceptors.

**Figure supplement 2.** Immunofluorescence staining of retinal enzymes and transporters in vivo and in vitro.

component analysis showed clear group separation (*Figure 1I*). Unbiased clustering of metabolite profiles revealed specific groups that were differentiated among the five experimental situations (*Figure 2A*). The greatest cluster overlap was seen in control and Shikonin groups. Significant clustering was observed in several amino acid sub-classes and energy metabolites. We found the strongest changes in the 1,9-DDF condition, where glutamate, aspartate, alanine, *O*-phosphocholine, and nicotinamide adenine dinucleotide (NAD$^+$) concentrations were particularly upregulated compared to control. Further, branched chain amino acids (BCAAs), lactate, and threonine were upregulated in the FCCP and *no* RPE condition. Glutathione (GSH) and glutathione disulfide (GSSG) were relatively reduced by Shikonin treatment, with increased creatine and formate. Control and 1,9-DDF treatment displayed high levels of guanosine triphosphate (GTP), ATP, and π-methylhistidine, which were low in the other groups (*Figure 2A*).

We then compared metabolite patterns for each treatment to control (*Figure 2B*). Metabolites were grouped according to class and function, including fundamental metabolic pathways, such as energy and cofactor metabolism, coenzymes, Krebs-cycle, neuronal tissue-associated metabolites and neurotransmitters, BCAAs, glucogenic and ketogenic amino acids (AAs), serine synthesis pathway, cell membrane synthesis, and individual features from redox metabolism.

To get an overall impression of treatment effects on energy metabolism, we assessed the adenylate energy charge of the retina under various experimental conditions and the ratio of GSH to GSSG to investigate oxidative/reductive stress. The energy charge appeared to be reduced in the Shikonin and FCCP condition, while the GSH/GSSG ratio of the tissue was not significantly changed from control in any condition (*Figure 2—figure supplement 3A and B*). We then analysed the spent culture medium at post-natal day (P) 15 to assess retinal glucose consumption and lactate production after 2 days of culture. As could be expected, 1,9-DDF treatment reduced glucose consumption compared to control, while FCCP treatment increased glucose expenditure (*Figure 2—figure supplement 2C*). Conversely, lactate production was decreased in the 1,9-DDF group. However, in the *no* RPE condition and after FCCP treatment lactate concentrations in the spent medium were strongly and significantly elevated (*Figure 2—figure supplement 2D*).

Afterwards, the metabolomic differences identified within the retina were explored in detail and related to the retinal expression of corresponding enzymes.

## Retina cultured without RPE displays strongly increased glycolytic activity

The RPE is tightly bound to the choroid and serves as interface between vasculature and neuroretina. Because of the strong adherence of RPE to the choroid, most prior studies on explanted retina were performed without RPE (*Warburg, 1925*; *Kanow et al., 2017*). To understand the metabolic communication between neuroretina and RPE, we prepared organotypic retinal explants both with RPE (control) or without RPE (*no* RPE). Since most previous studies on retinal energy metabolism had used retina without RPE (*Warburg, 1925*; *Winkler, 1981*), this comparison also allowed us to relate our data to those earlier results.

The metabolite profile of control was markedly different from retina cultured without RPE, with 25 metabolites exhibiting significant changes (*Figure 3A*, *Figure 3—figure supplement 1*). Decreased ATP levels in the *no* RPE group corresponded to increased lactate. The BCAAs isoleucine, leucine, and valine, and some of the glucogenic/ketogenic AAs threonine, methionine, glycine, displayed higher levels in the *no* RPE group compared to control. Alanine, aspartate, and taurine showed the opposite trend. Hypotaurine, 4-aminobutyrate (GABA), and glutamate were accumulated in retinal tissue, while

**Table 1.** Primary and secondary antibodies used in the study, providers, and dilutions.

| Reagent or resource | Source | Identifier |
|---|---|---|
| Antibodies, dilution | | |
| Mouse monoclonal anti-RPE65, 1:100 | Thermo Fisher Scientific | Cat# MA5-16042, RRID:AB_11151857 |
| Rabbit polyclonal anti-GLUT1, 1:300 | Abcam | Cat# ab652, RRID:AB_305540 |
| Rabbit polyclonal anti-GLUT3, 1:300 | Abcam | Cat# ab41525, RRID:AB_732609 |
| Rabbit monoclonal anti-PKM1, 1:300 | Cell Signaling Technology | Cat# 7067, RRID:AB_2715534 |
| Rabbit monoclonal anti-PKM2, 1:300 | Cell Signaling Technology | Cat# 4053, RRID:AB_1904096 |
| Mouse monoclonal anti-Cytochrome C, 1:300 | Molecular Probes | Cat# A-6403, RRID:AB_221582 |
| Rabbit polyclonal anti-Cone arrestin, 1:300 | Sigma-Aldrich | Cat# AB15282; RRID:AB_1163387 |
| Rabbit polyclonal anti-Citrate synthase, 1:300 | GeneTex | Cat# GTX110624, RRID:AB_1950045 |
| Rabbit polyclonal anti-ATP synthase gamma, 1:300 | GeneTex | Cat# GTX114275S, RRID:AB_10726795 |
| Rabbit polyclonal anti-Fumarate hydratase, 1:300 | GeneTex | Cat# GTX109877, RRID:AB_1950283 |
| Rabbit polyclonal anti-Taurine, 1:300 | Abcam | Cat# ab9448, RRID:AB_307261 |
| Rabbit monoclonal anti-Alanine transaminase, 1:300 | Abcam | Cat# ab202083, RRID:AB_2915976 |
| Rabbit polyclonal anti-SUCLG-1, 1:300 | Novus Biologicals | Cat# NBP1-32728, RRID:AB_2286802 AB_2286802 |
| Mouse monoclonal anti-Glutamine synthetase, 1:1000 | Abcam | Cat# MAB302, RRID:AB_2110656 |
| Rabbit polyclonal anti-Glutaminase C (GAC), 1:300 | GeneTex | Cat# GTX131263, RRID:AB_2886452 |
| Rabbit polyclonal anti-PCK1, 1:300 | Affinity Biosciences | Cat# DF6770, RRID:AB_2838732 |
| Rabbit monoclonal anti-PCK2, 1:300 | Novus Biologicals | Cat# NBP2-75610 RRID:AB_2915974 |
| Peanut agglutinin (PNA), 1:1000 | Vector Laboratories | Cat# FL-1071, RRID:AB_2315097 |
| Rabbit polyclonal anti-Aspartate aminotransferase-1, 1:300 | Abcam | Cat# ab221939, RRID:AB_2915980 |
| Rabbit polyclonal anti-FABP-1 (Aspartate aminotransferase-2), 1:300 | Abcam | Cat# ab153924, RRID:AB_2915981 |
| Alexa Fluor 568, goat anti-mouse, 1:500 | Molecular Probes | Cat# 11031, RRID:AB_144696 |
| Alexa Fluor 568, goat anti-rabbit, 1:500 | Molecular Probes | Cat# 11036, RRID:AB_10563566 |
| Alexa Fluor 568, donkey anti-rabbit, 1:500 | Invitrogen | Cat# 10042, RRID:AB_2534017 |
| Alexa Fluor, 488 donkey anti-mouse, 1:500 | Thermo Fisher | Cat# 21206, RRID:AB_2535792 |
| Alexa Fluor 488, donkey anti-rabbit, 1:500 | Thermo Fisher | Cat# A21206, RRID:AB_2535792 |
| Alexa Fluor 488, goat anti-rabbit, 1:500 | Molecular Probes | Cat# A11034, RRID:AB_2576217 |

membrane synthesis-associated metabolites, such as myo-inositol and *O*-phosphocholine, as well as creatine and *N*-acetylaspartate (NAA) were reduced in neuroretina. Moreover, higher levels of GSSG appeared in the absence of RPE, pointing towards an altered redox metabolism.

Pattern hunter (Pearson r) correlation analysis found citrate, a central Krebs-cycle metabolite, positively correlated with taurine, alanine, aspartate, and ATP (*Figure 3B*). Negative correlations with

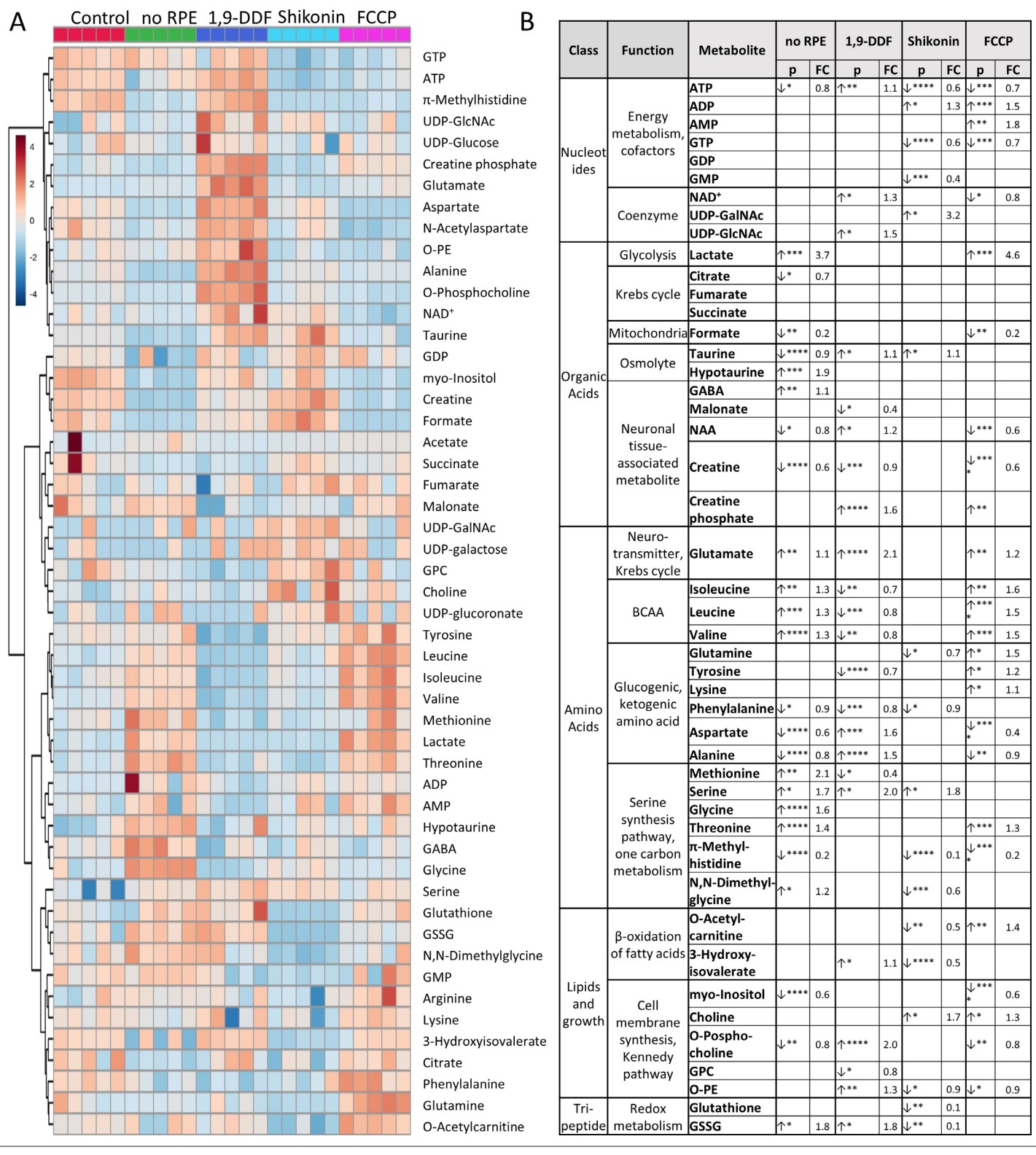

**Figure 2.** ¹H-NMR spectroscopy-based metabolomics analysis of retina subjected to interventions in energy metabolism. (**A**) Heatmap based on unsupervised hierarchical cluster analysis by Ward's linkage, showing auto-scaled metabolite concentrations (red – high, blue – low) in five different experimental conditions: control – red, *no* retinal pigment epithelium (RPE) – green, 1,9-dideoxyforskolin (1,9-DDF) – dark blue, Shikonin – light blue, carbonyl cyanide-*p*-trifluoromethoxyphenylhydrazone (FCCP) purple (n=5 samples per condition). Squares in the heatmap represent data from individual retinal explants from different animals (n=5). (**B**) Metabolic profiles of each intervention were compared to control. Metabolites significantly

*Figure 2 continued on next page*

*Figure 2 continued*

changed in at least one experimental condition were grouped according to functions and pathways. Data show p-values and fold change (FC) over control. Statistical comparison: Student's unpaired t-test (group variance equal); p-values: ****<0.0001, ***<0.001, **<0.01, *<0.05.

The online version of this article includes the following source data and figure supplement(s) for figure 2:

**Source data 1.** Full list of all quantified metabolites in retina samples.

**Figure supplement 1.** Retina characterisation by metabolomics and immunostaining.

**Figure supplement 2.** Retinal tissue energy status, glutathione (GSH) ratio, glucose, and lactate concentrations in medium.

citrate included ADP, BCAAs, and lactate. A subsequent Kyoto Encyclopaedia of Genes and Genomes (KEGG)-based pathway analysis revealed changes in arginine and proline metabolism, taurine and hypotaurine metabolism, and alanine, aspartate, and glutamate metabolism in the *no* RPE situation (*Figure 3C*).

Immunofluorescence was used to localise enzymes likely responsible for the observed metabolic changes (*Figure 3D*). Citrate synthase (CS) and fumarase, key enzymes of the Krebs-cycle, were found mostly in photoreceptor inner segments and synaptic terminals. The γ-subunit of ATP synthase, essential for OXPHOS, was also localised to photoreceptor inner segments and synapses. Thus, low ATP levels in the *no* RPE group likely resulted from decreased photoreceptor Krebs-cycle and OXPHOS activity. High levels of taurine were detected in photoreceptor inner segments and synapses, as well as in the ganglion cell layer (*Figure 3D*). Finally, the enzyme alanine transaminase (ALT), previously thought to be expressed only in muscle and liver, was found to be expressed in the inner nuclear and plexiform layers (*Figure 3D*). ALT was also strongly expressed in cone inner segments, as evidenced by co-labelling with cone-Arr3. This expression pattern and high alanine levels in the control strongly suggested Cahill-cycle activity in the retina (*Felig, 1973*). When ALT activity was inhibited with β-chloro-alanine, photoreceptor cell death was increased and notably cone survival was reduced, functionally validating the expression of ALT in the outer retina (*Figure 3—figure supplement 2*).

The patterns observed in the *no* RPE to control comparison indicated that retinal metabolism strongly depended on RPE-to-neuroretina interactions, including shuttling of metabolites such as hypotaurine/taurine. Notably, the accumulation of BCAA, glutamate, and lactate, concomitant with decreased alanine levels in the *no* RPE group, implied that RPE removal switched retinal metabolism from Krebs-cycle/OXPHOS to aerobic glycolysis.

## Photoreceptors use the Krebs-cycle to produce GTP

Since in *Figure 2*, the *no* RPE and FCCP groups showed similar metabolite patterns and pathway changes overall, we subjected these two experimental groups to a detailed statistical analysis. This revealed 17 significant metabolite changes (*Figure 4A*; *Figure 4—figure supplement 1*). Among the most highly reduced metabolite concentrations in the FCCP group was GTP, which positively correlated with GABA, ATP, GSSG, hypotaurine, and $NAD^+$ (*Figure 4B*). In contrast, GDP, BCAAs, glutamine, and citrate negatively correlated to GTP. The KEGG-based pathway analysis ranked glycine, serine, and threonine metabolism, GSH metabolism, and alanine, aspartate, and glutamate metabolism as the most significantly changed between the two conditions (*Figure 4C*).

Curiously, measurable amounts of GTP can only be generated in two ways: (1) by succinate-CoA ligase, GTP-forming-1 (SUCLG-1) in the Krebs-cycle, and (2) from excess ATP by nucleoside-diphosphate kinase, an enzyme expressed in photoreceptor inner and outer segments (*Abdulaev et al., 1998*; *Rueda et al., 2016*). Since ATP levels were low in both the FCCP and the *no* RPE situation, here, GTP could not have been produced from ATP. Immunofluorescence for SUCLG-1 showed strong expression in photoreceptor inner segments and synapses, where it co-localised to a large extent with mitochondrial COX (*Figure 4D*). Hence, the SUCLG-1 retinal expression pattern and the synthesis of GTP in the absence of RPE provided clear evidence for Krebs-cycle activity in photoreceptors.

We also identified an FCCP-induced upregulation of citrate, concomitant with a downregulation of $NAD^+$. This indicated a Krebs-cycle interruption at the level of D-isocitrate to α-ketoglutarate conversion, i.e., a step that requires $NAD^+$. $NAD^+$ is known to be consumed by poly-ADP-ribose-polymerase (PARP), which is activated by oxidative DNA damage (*Bai, 2015*) and upregulated in dying photoreceptors (*Paquet-Durand et al., 2007*). Therefore, we quantified poly-ADP-ribose (PAR) accumulation in retinal cells as a marker for PARP activity. Compared to the *no* RPE group, more photoreceptor cells

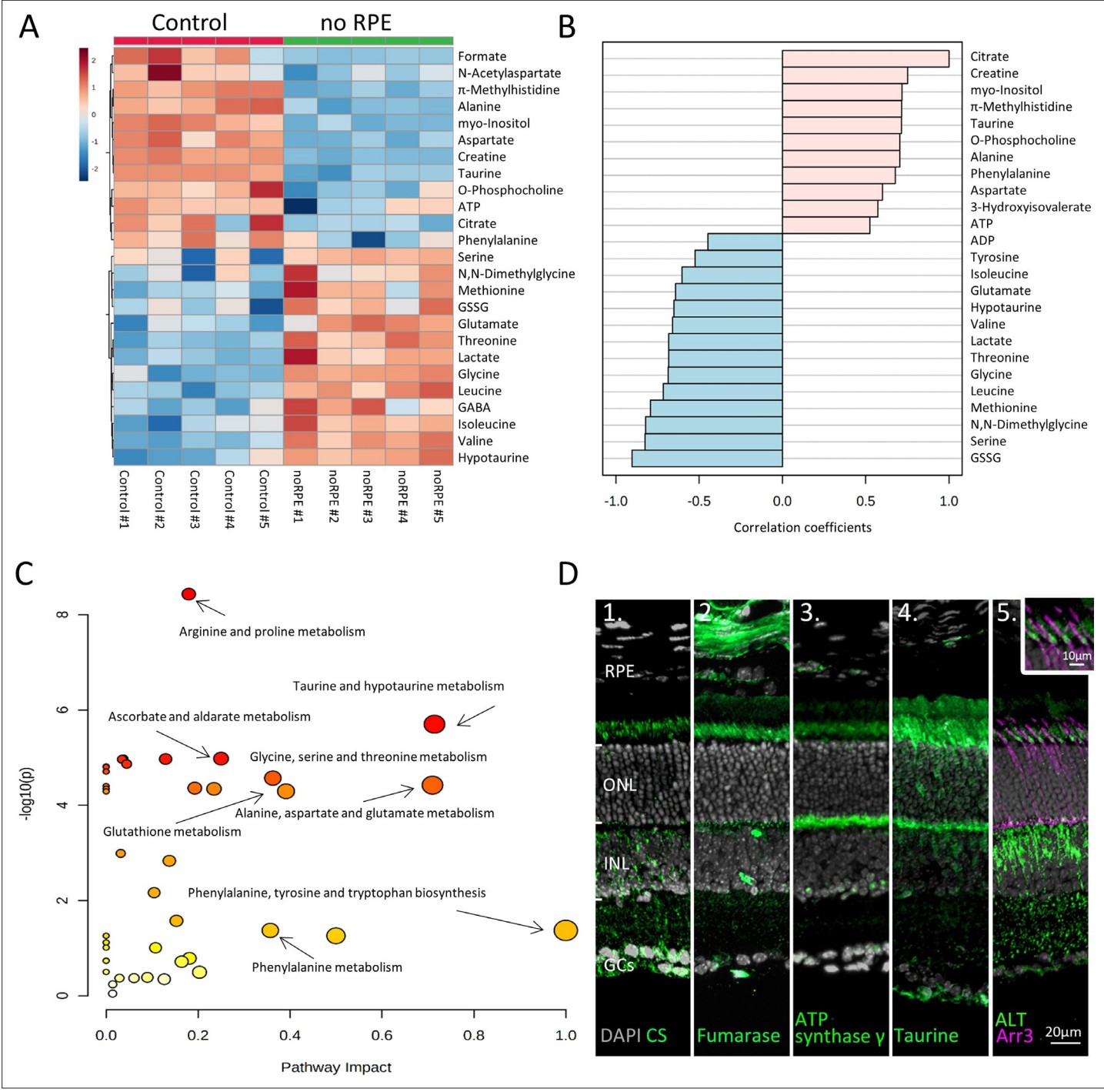

**Figure 3.** Absence of retinal pigment epithelium (RPE) dramatically changes retinal metabolism. (**A**) Heatmap, based on unsupervised hierarchical cluster analysis by Ward's linkage, illustrating statistically significant changes for 25 metabolites (n=5; unpaired t-test, fold change [FC] >1.2, raw p-value<0.05). (**B**) Pattern hunter for citrate, showing the top 25 correlating metabolites, correlation coefficient given as Pearson r distance. (**C**) Metabolic pathways most affected in *no* RPE condition compared to control, based on Kyoto Encyclopaedia of Genes and Genomes (KEGG) pathway database. (**D**) Immunodetection of enzymes and metabolites (green) on in vivo retina. (**D1**) Citrate synthase (CS), (**D2**) fumarase, (**D3**), adenosine triphosphate (ATP) synthase γ, (**D4**) taurine, (**D5**) alanine transaminase (ALT). Co-staining for ALT and cone arrestin (Arr3; magenta) showed high expression in cone inner segments. DAPI (grey) was used as a nuclear counterstain.

The online version of this article includes the following figure supplement(s) for figure 3:

**Figure supplement 1.** Metabolomic comparison of no retinal pigment epithelial (RPE) condition vs. control.

**Figure supplement 2.** Inhibition of alanine transaminase (ALT) and glutaminase C (GAC) dose-dependently reduce photoreceptor viability.

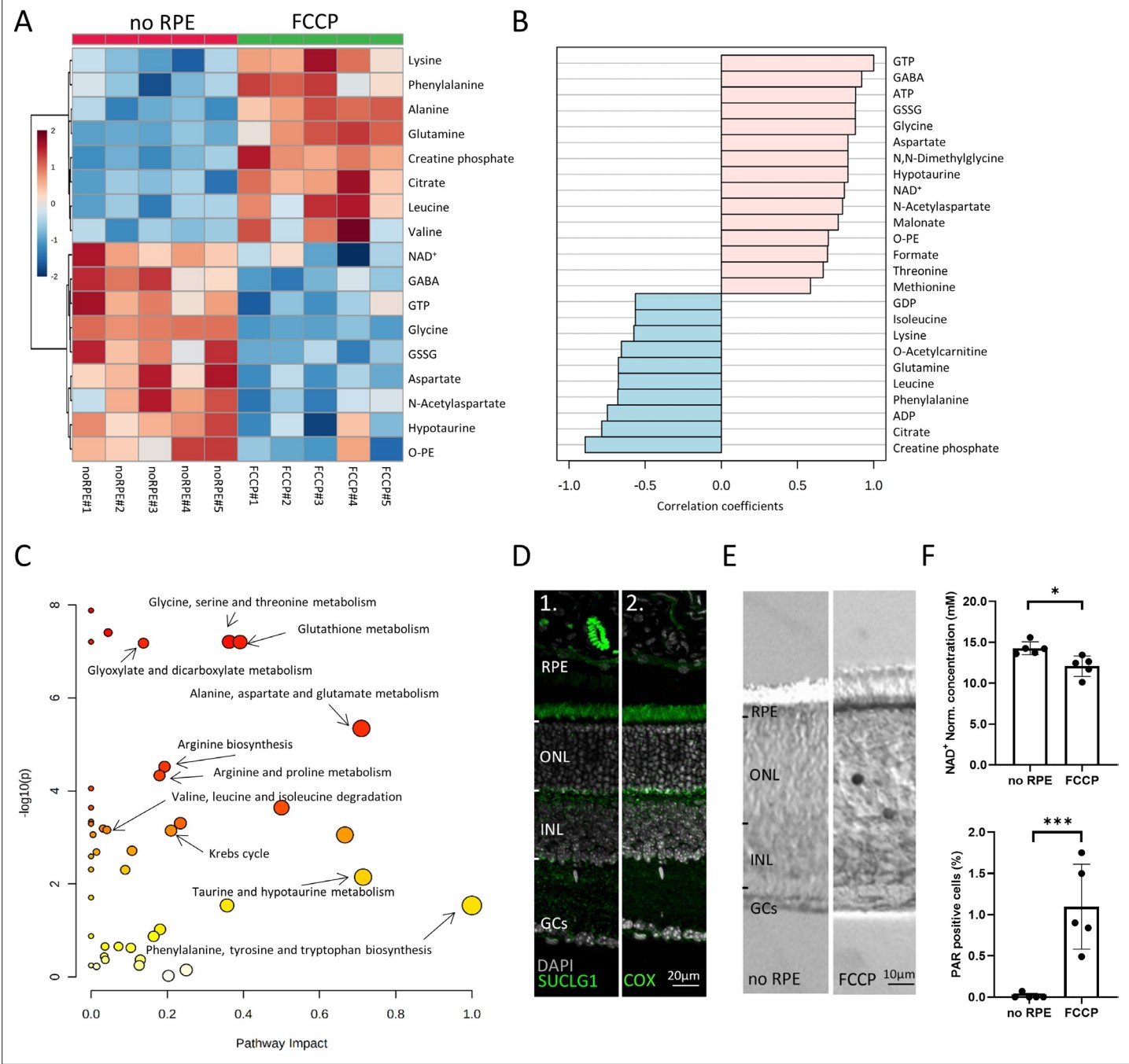

**Figure 4.** Comparison between *no* retinal pigment epithelial (RPE) and carbonyl cyanide-*p*-trifluoromethoxyphenylhydrazone (FCCP) conditions. (**A**) Heatmap, based on unsupervised hierarchical cluster analysis by Ward's linkage, illustrating statistically significant metabolite changes (n=5; unpaired t-test, fold change >1.2, raw p-value<0.05). (**B**) Pattern hunter for guanosine triphosphate (GTP) showing the top 25 correlating compounds. Correlation coefficient given as Pearson r distance. (**C**) Kyoto Encyclopaedia of Genes and Genomes (KEGG)-based pathway analysis, comparison between *no* RPE and FCCP. (**D**) Immunofluorescence for succinate-CoA ligase-1 (SUCLG-1, green) labelled photoreceptor inner segments and co-localised with cytochrome oxidase (COX). DAPI (grey) was used as a nuclear counterstain. (**E**) PAR-positive photoreceptors (black) in the outer nuclear layer (ONL) of FCCP-treated retina. (**F**) Compared to *no* RPE, NAD+ levels were lower in the FCCP group, while the percentage of PAR-positive cells was higher. Data represented as mean ± SD (n=5). Statistical testing: Student's two-tailed t-test. p-Values: ***<0.001, *<0.05.

The online version of this article includes the following figure supplement(s) for figure 4:

**Figure supplement 1.** Metabolomic comparison of no retinal pigment epithelial (RPE) vs. carbonyl cyanide-*p*-trifluoromethoxyphenylhydrazone (FCCP) treatment.

in the FCCP-treated retina showed PAR accumulation, correlating with decreased retinal NAD$^+$ levels (*Figure 4E and F*). Hence, FCCP-induced oxidative stress may cause increased PARP activity and decreased NAD$^+$ levels, eventually interrupting the Krebs-cycle, as evidenced by citrate accumulation.

Overall, the data from the *no* RPE to FCCP group comparison showed that disruption of OXPHOS led to AA accumulation, including lysine, phenylalanine, glutamine, leucine, and valine. Notably, alanine accumulation in the FCCP group was likely caused by ALT-dependent pyruvate transamination. By contrast, metabolites high in the *no RPE* group but low with FCCP treatment were probably related to Krebs-cycle in the neuroretina. This concerned especially GTP, aspartate, and NAA.

### Reduced retinal glucose uptake promotes anaplerotic metabolism

Since GLUT1 was predominantly expressed in the RPE, we initially assumed that the metabolic response to 1,9-DDF treatment might resemble that of the *no* RPE situation. However, the patterns found between these two groups were very different (*Figure 2*) and to explore these differences further, we performed an additional two-way comparison. This analysis revealed 28 significantly changed metabolites (*Figure 5A*; *Figure 5—figure supplement 1A*). When compared to the *no* RPE, the 1,9-DDF group showed changes in mitochondria and Krebs-cycle-associated metabolites (e.g. high ATP, high formate, low BCAA). Conversely, the Cahill-cycle product alanine and NAA were upregulated by 1,9-DDF treatment. The pattern hunter analysis showed that metabolites formate, glutamate, taurine, and citrate were positively correlated with ATP, while, AMP, GMP, and BCAA were negatively correlated with ATP (*Figure 5B*).

The KEGG pathway analysis showed alanine, aspartate, and glutamate metabolism, GSH metabolism and glycine, serine, and threonine metabolism among the three most affected pathways (*Figure 5—figure supplement 1B*). Taken together, the depletion of BCAAs in the 1,9-DDF treatment indicated that either the RPE, the neuroretina, or both used anaplerotic substrates to fuel the Krebs-cycle and maintain ATP production.

### Inhibition of glycolysis strongly impacts retinal ATP production

Next, we compared the Shikonin treatment, which inhibits the last step of glycolysis, to the FCCP treatment, which abolishes mitochondrial ATP synthesis, i.e., we compared two experimental situations that could be seen as manipulations of the two opposing ends of energy metabolism, non-oxidative glycolysis and OXPHOS. This analysis showed increased formate, aspartate, and NAA levels in the Shikonin group (*Figure 5C*). Conversely, higher amounts of GSH, GSSG, BCAA, threonine, and phenylalanine after FCCP treatment indicated Krebs-cycle interruption (*Figure 5—figure supplement 2A*). The production of lactate was higher with FCCP compared to Shikonin treatment, probably reflecting increased glycolysis to compensate for the loss of Krebs-cycle-dependent ATP production. Metabolites connected to oxidative stress, such as GSSG and GSH, were also increased by FCCP treatment. Pattern hunter for metabolites correlated with GSH found 3-hydroxyisovalerate, glutamine, GSSG, but also lactate and ATP, while succinate, alanine, and GABA were negatively correlated with GSH (*Figure 5D*). The subsequent pathway analysis identified arginine biosynthesis, alanine, aspartate, and glutamate metabolism, as well as glycolysis and gluconeogenesis as the most strongly regulated metabolic pathways (*Figure 5—figure supplement 2B*).

### Blocking glucose uptake and OXPHOS disruption reveal distinct rod and cone metabolism

The metabolic rate of cones is at least two times higher than that of rods (*Ingram et al., 2020*). The treatment with 1,9-DDF resulted in near-complete cone loss, while cones were fully preserved with FCCP treatment (*Figure 1*). Although we did not expect these differential treatment effects on cone survival, we used this outcome in an explorative approach to try and investigate the relative contributions of rods and cones to the metabolite patterns observed.

In this three-way comparison, 29 metabolite concentrations were significantly changed (*Figure 6A*; *Figure 6—figure supplement 1*). The FCCP treatment reduced ATP and GTP levels, while lactate production was greatly increased, along with methionine and threonine. BCAAs, glutamine, phenylalanine, and tyrosine also showed a significant upregulation with FCCP treatment. Interestingly, some metabolites like GSSG, serine, glutamate, alanine, NAD$^+$, and taurine displayed a pronounced increase in the 1,9-DDF group when compared to either control or FCCP. In addition, formate, aspartate, and

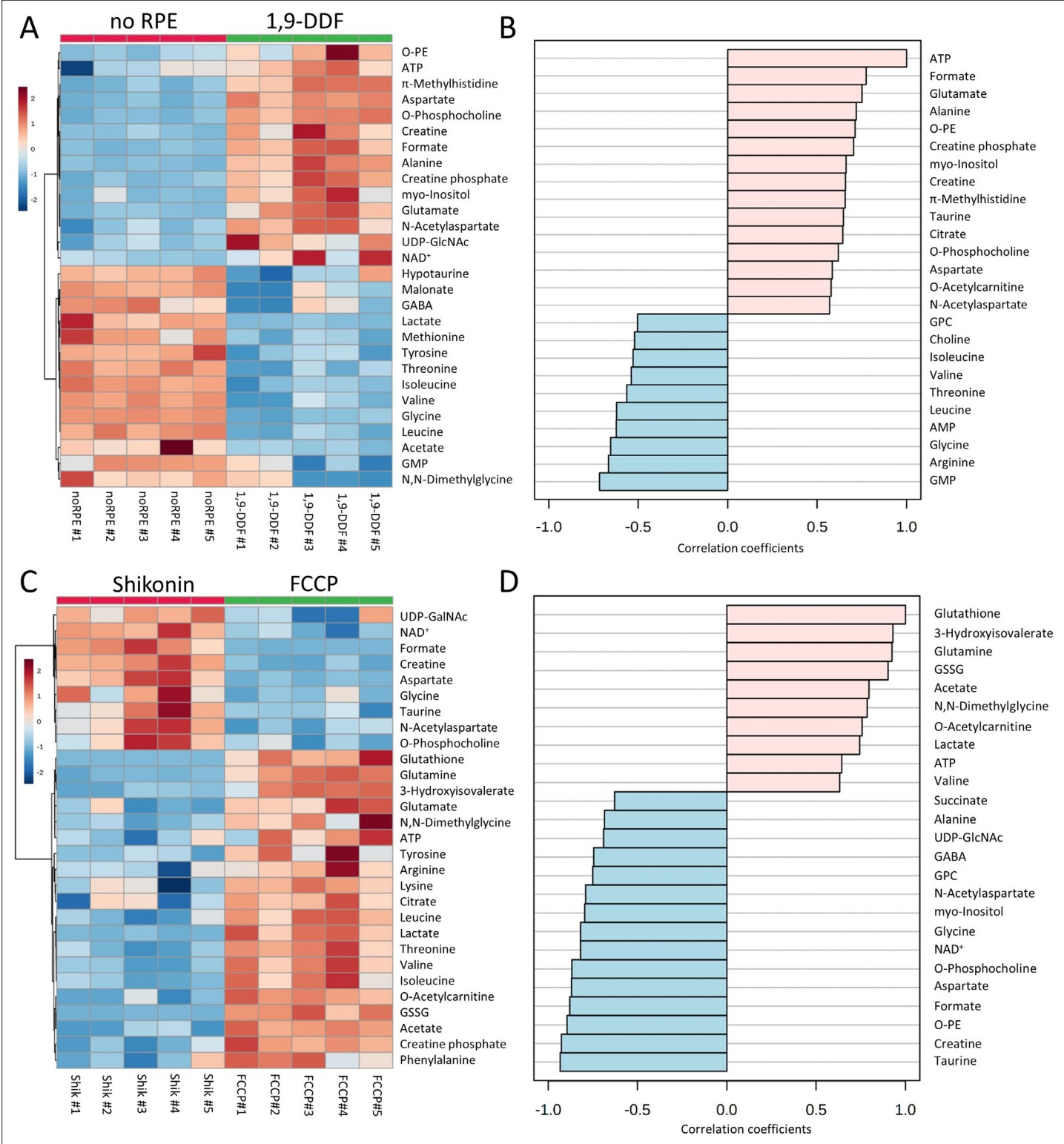

**Figure 5.** Metabolomic analysis of *no* retinal pigment epithelial (RPE) vs.1,9-dideoxyforskolin (1,9-DDF) treatment and Shikonin vs. carbonyl cyanide-*p*-trifluoromethoxyphenylhydrazone (FCCP) treatment. Heatmap illustrating statistically significant metabolite changes (n=5; unpaired t-test, fold change [FC] >1.2, raw p-value<0.05) for (**A**) *no* RPE vs. 1,9-DDF and (**C**) Shikonin vs. FCCP. Pattern hunter for (**B**) adenosine triphosphate (ATP) and (**D**) glutathione, showing the top 25 correlating compounds. Correlation coefficient given as Pearson r distance.

The online version of this article includes the following figure supplement(s) for figure 5:

*Figure 5 continued on next page*

*Figure 5 continued*

**Figure supplement 1.** Metabolomic comparison of no retinal pigment epithelial (RPE) vs.1,9-dideoxyforskolin (1,9-DDF) treatment.

**Figure supplement 2.** Metabolomic comparison of carbonyl cyanide-*p*-trifluoromethoxyphenylhydrazone (FCCP) vs. Shikonin treatments.

myo-inositol were upregulated with 1,9-DDF treatment, together with NAA and creatine. Finally, the comparison between the 1,9-DDF and FCCP groups revealed highly increased levels of ATP and GTP, implying a strong activation of the Krebs-cycle under 1,9-DDF treatment.

A cell type-specific attribution of these metabolite patterns may be superseded by direct drug treatment effects. Using only metabolite changes in opposing directions from control to interpret different cellular compositions, we identified BCAAs, tyrosine, NAA, and ATP (*Figure 6—figure supplement 1*). BCAAs and tyrosine (and glutamine) exhibited low levels in the rod-rich 1,9-DDF group and high levels in the cone-rich FCCP group, indicating that these metabolites might be consumed by rods or produced in cones. Vice versa, it may be assumed that aspartate and NAA were predominantly produced in rods.

To investigate the role of glutamine in retinal metabolism, we used immunofluorescence (*Figure 6B*). We confirmed a strong expression of glutamine synthase (GS) in MGCs (*Riepe and Norenburg, 1977*). Remarkably, the enzyme that hydrolyses glutamine to glutamate, glutaminase C (GAC) was prominently expressed in photoreceptor inner segments, photoreceptor synapses, and INL. Co-labelling with cone Arr3 revealed a strong expression of GAC in cone inner segments, implying that cones could use glutamine as a metabolic substrate. Accordingly, treatment with the GAC-specific inhibitor Compound 968 dose-dependently reduced photoreceptor viability – affecting cones more strongly than rods – while leaving the inner retina mostly intact (*Figure 3—figure supplement 2C and D*).

## Impact on glycolytic activity, serine synthesis pathway, and anaplerotic metabolism

To assess how the various experimental interventions affected retinal metabolism, we calculated the ratios for pathway-specific metabolites (*Figure 6C*). GTP, when produced by SUCLG-1, is a marker for Krebs-cycle activity, while lactate indicates glycolytic activity. In the *no* RPE group, the GTP to lactate ratio was significantly lower than in control, indicating 4.1 times higher glycolytic activity. While in the 1,9-DDF and Shikonin group the GTP to lactate ratios were not significantly different from control, with FCCP treatment this ratio dropped to the lowest level, in line with a strong downregulation of Krebs-cycle and concomitant upregulation of glycolysis.

We then calculated the ratio of serine to lactate as an indicator for serine synthesis pathway activity. Under control conditions, and in the *no* RPE and FCCP groups, this activity was rather low, while it was strongly increased by 1,9-DDF and Shikonin treatment. In the Shikonin group, serine production from 3-phosphoglycerate and subsequent deamination to pyruvate may bypass the PKM2 block. Serine is also a precursor of phosphatidylserine, one of the three main cell membrane components. Together with a reduction of choline and high concentrations of myo-inositol, *O*-phosphocholine, and *O*-phosphoethanolamine, this may reflect high cell membrane synthesis under 1,9-DDF treatment.

Finally, we used the ratio of GTP to BCAAs to investigate to what extent Krebs-cycle activity was driven by anaplerotic metabolism. A comparatively low GTP to BCAA ratio in the *no* RPE situation showed that much of the GTP generated in the neuroretina likely came from anaplerotic substrates. While both Shikonin and FCCP treatment decreased BCAA use, 1,9-DDF treatment increased anaplerotic metabolism.

## Evidence for mini-Krebs-cycle activity

Comparing the ratios of GTP to lactate with GTP to BCAA in the *no* RPE situation showed that a large proportion of the GTP produced in the neuroretina by SUCLG-1 was derived from anaplerotic substrates rather than from pyruvate. These substrates enter the Krebs-cycle mostly at the level of α-ketoglutarate, suggesting that the neuroretinal Krebs-cycle might not start with citrate. Instead, only an α-ketoglutarate to oxaloacetate shunt would be employed, including the GTP-synthesising step from succinyl-CoA to succinate (*Figure 7A*). A key step of this mini-Krebs-cycle (*Riepe and Norenburg, 1977*) is the transamination of oxaloacetate with glutamate by aspartate aminotransferase (AST) to yield α-ketoglutarate and aspartate, which is further acetylated to NAA. To investigate this possibility,

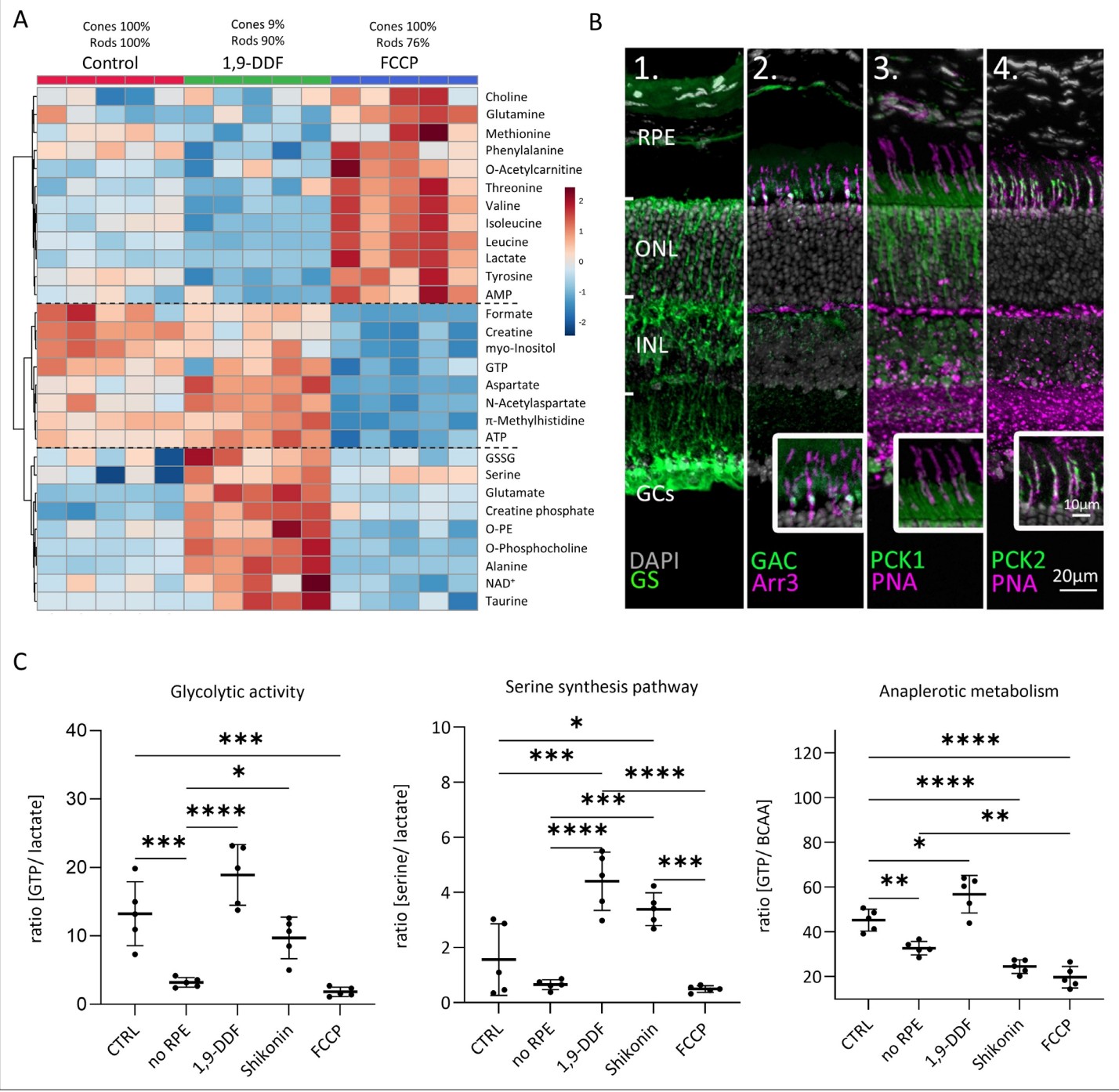

**Figure 6.** Metabolomic comparison between control, 1,9-dideoxyforskolin (1,9-DDF), and carbonyl cyanide-*p*-trifluoromethoxyphenylhydrazone (FCCP) treatment. (**A**) Heatmap illustrating 29 statistically significant metabolite changes (parametric one-way ANOVA, Fisher's LSD post hoc analysis). Three main clusters of metabolite changes were evident (dashed lines). (**B**) Immunostaining (green) for enzymes related to glutamine metabolism. DAPI (grey) was used as nuclear counterstain. (**B1**) Glutamine synthase (GS), (**B2**) glutaminase C (GAC); co-localisation with cone-arrestin (Arr3; magenta). (**B3**) Phosphoenolpyruvate carboxykinase 1 (PCK1) did not co-localise with cone marker peanut agglutinin (PNA) while (**B4**) PCK2 did. (**C**) Ratios between metabolites representing glycolysis (lactate vs. guanosine triphosphate [GTP]), anaplerotic metabolism (GTP vs. branched chain amino acids [BCAA]), and serine synthesis pathway (serine vs. lactate). Data represented as individual data points with mean ± SD (n=5). Statistical testing: one-way ANOVA with Tukey's post hoc test; p-values: ****<0.0001, ***<0.001, **<0.01, *<0.05.

The online version of this article includes the following figure supplement(s) for figure 6:

**Figure supplement 1.** Metabolomics analysis of control vs. 1,9-dideoxyforskolin (1,9-DDF) vs. carbonyl cyanide-*p*-trifluoromethoxyphenylhydrazone (FCCP) three-way comparison.

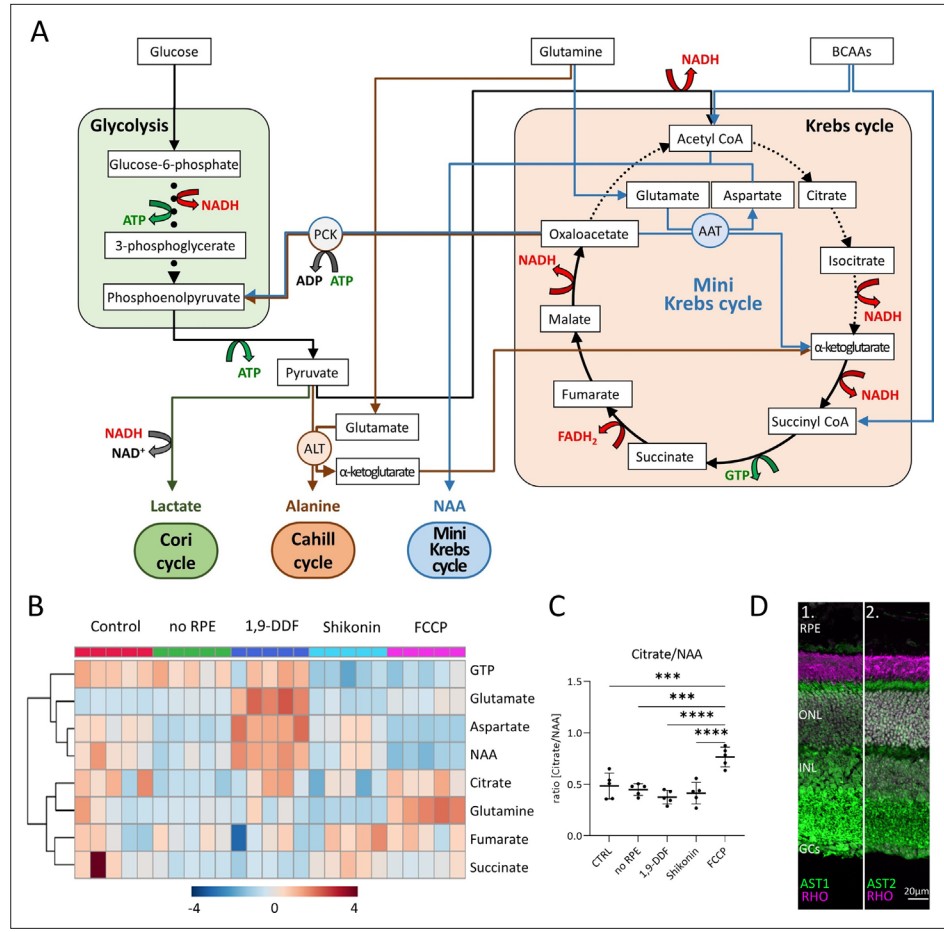

**Figure 7.** Metabolic pathways in the retina, key metabolites, and expression of aspartate aminotransferase (AST). (**A**) Overview of main metabolic pathways and metabolites. Execution of Cori- (green arrows), Cahill- (brown), or mini-Krebs-cycle (blue) releases the signature metabolites lactate, alanine, and *N*-acetylaspartate (NAA), respectively. Key enzymes of the Cahill- and mini-Krebs-cycle are phosphoenolpyruvate carboxykinase (PCK), pyruvate kinase M (PKM), alanine transaminase (ALT), and AST. (**B**) Hierarchical clustering of eight metabolites connected to Krebs and mini-Krebs cycle. (**C**) Ratio of NAA vs. citrate, representing full and mini-Krebs-cycle. Data represented as ratio of individual data points with mean ± SD (n=5). Statistical comparison using one-way ANOVA, Tukey's multiple comparisons test; p-values: ****<0.0001, ***<0.001. (**D**) Immunostaining for aspartate aminotransferase-1 and -2 (AST1, AST2; green) co-stained with the rod photoreceptor outer segment marker rhodopsin (RHO; magenta), DAPI (grey) was used as nuclear counterstain.

The online version of this article includes the following figure supplement(s) for figure 7:

**Figure supplement 1.** Proposed metabolic interactions between different retinal cell types.

we analysed eight metabolites associated with the Krebs-cycle (citrate, succinate, fumarate, GTP) and the hypothesised mini-Krebs-cycle (glutamine, glutamate, aspartate, NAA).

Hierarchical clustering showed similar patterns for aspartate and NAA in all experimental conditions (*Figure 7B*). Except for the FCCP treatment, the ratio of citrate/NAA was 0.5, indicating that the retina preferred the mini-Krebs-cycle over the full Krebs-cycle (*Figure 7C*). Immunolabelling for AST – conventionally associated with muscle and liver metabolism (*Nathwani et al., 2005*; *Kim et al., 2008*) – found both AST1 and AST2 to be expressed in photoreceptor inner segments and cell bodies (*Figure 7D*). This confirmed that photoreceptors can execute the mini-Krebs-cycle, while the NAA production seen in the metabolomic data suggested that this cycle was indeed used (*Figure 7— figure supplement 1*).

Finally, we compared the energetic efficiencies of glycolysis, Krebs-, Cori-, Cahill-, and mini-Krebs cycle (*Figure 8*). Using anaplerotic substrates the Cahill- and mini-Krebs cycles allow for far more efficient energy production compared to the Cori-cycle. Moreover, mitochondrial PCK2 can regenerate

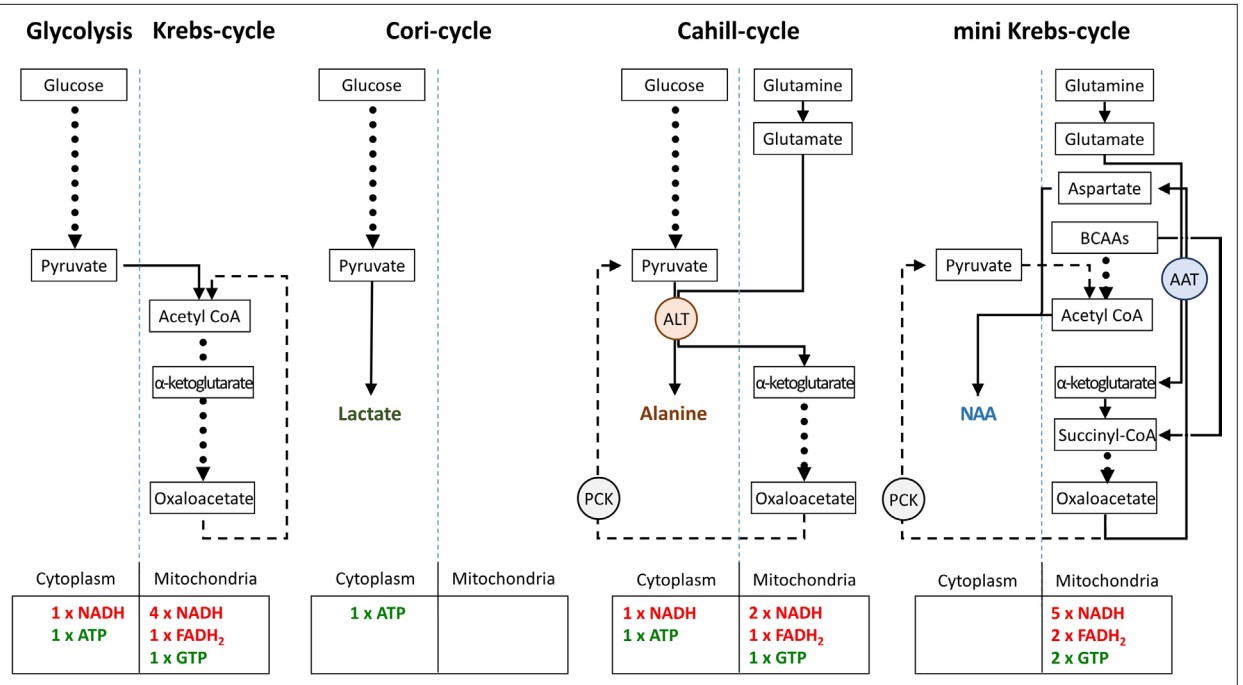

**Figure 8.** Comparison of metabolic pathways and their energetic efficiencies. Compared to the Cori-cycle, both Cahill- and mini-Krebs-cycles are highly efficient. Their key enzymes – alanine transaminase (ALT) and aspartate aminotransferase (AST), respectively – generate alanine and aspartate/*N*-acetylaspartate (NAA). Pyruvate carboxy kinase (PCK), either in cytoplasm or within mitochondria, may reconstitute pyruvate from oxalacetate. Note that energy output of each pathway was calculated based on input of pyruvate (three carbons) or glutamate/branched chain amino acid/acetyl-CoA (three carbons).

pyruvate from oxaloacetate. Assuming that 1 mole of NADH/FADH$_2$ can be used to generate 3 moles of ATP via OXPHOS, and depending on the exact stoichiometry of glutamate/BCAA input, the mini-Krebs-cycle may deliver up to 18 moles of ATP and 2 moles of GTP per 3 carbon unit. Astonishingly, this exceeds the 15 moles of ATP and 1 mole of GTP generated in the 'full' Krebs-cycle.

## Discussion

This study combined cellular enzyme expression patterns with quantitative metabolomics of retinal explants. In an entirely controlled environment, this enabled a comprehensive assessment of retinal metabolism and enabled an analysis of the impact of different experimental conditions on retinal cell death and survival. As opposed to earlier works, our study highlights the crucial importance of OXPHOS and anaplerotic pathways for the maintenance and survival of rod and cone photoreceptors. Importantly, we show that photoreceptors are able to uncouple glycolysis and Krebs-cycle using an NAA-producing shunt. This allows the use of both pathways in parallel and may resolve a longstanding problem in neuronal energy metabolism research. Because of the ramifications for overall cellular physiology, these findings are highly relevant for future therapy developments, for retinal diseases, as well as for neurodegenerative and further metabolic diseases in general.

### The retina as an experimental system for studies into neuronal energy metabolism

The study of neuronal metabolism in a living mammalian is notoriously difficult as there is constant interaction between different subcellular compartments, cell types, tissues, and organs (*Barros et al., 2023*). The retina may be a special case as it can function in isolation, e.g., as in an explant (*Haq et al., 2023*), which is why many prior studies on retinal metabolism employed explanted retina. However, in most retinal explant preparations the RPE separates from the neuroretina. Such explants were used in Otto Warburg's seminal studies in the 1920s, where the retina was found to release large amounts of lactate (*Warburg, 1925*). Later studies by Barry Winkler, using a similar experimental setup namely

explanted rat retina without RPE (*Winkler, 1981*), confirmed Warburg's original work and expanded on the idea that the neuroretina was using aerobic glycolysis.

In vivo studies on retinal metabolism are significantly more challenging and hence relatively rare. For example, in cats, the blood lactate concentration, sampled via choroidal vein cannulation, was compared against femoral artery lactate levels, concluding that a major fraction of retinal glucose was used for aerobic glycolysis (*Wang et al., 1997*). In contrast, monkey in vivo micro-electrode measurements indicated major oxygen consumption in photoreceptor inner segments, implying the use of OXPHOS (*Yu et al., 2005*). Future more comprehensive in vivo studies of retinal metabolism may require the simultaneous cannulation of central retinal artery and central retinal vein, combined with oxygen-sensing and metabolomic analysis. Such experiments may need to employ larger animal species (e.g. dog, monkey) to overcome limitations in eye and blood sample size.

Our in vitro approach may be seen as a compromise between earlier in vitro studies and the in vivo situation. Organotypic retinal explants maintain the histotypic context of the tissue, the interactions between its various cell types, full functionality, and light responsiveness, and are cultured under defined conditions, devoid of serum and antibiotics (*Haq et al., 2023*; *Tolone et al., 2023*). These retinal explants are derived from early post-natal retina (day 9) and can be cultured for up to 6 weeks in vitro without major cell loss (*Söderpalm et al., 1994*). Importantly, our retinal explant cultures include the RPE (except for the *no* RPE condition) and may thus be closer to the in vivo situation than previous in vitro studies. While the combination of this in vitro approach with [1]H-NMR metabolomics can be very powerful, it assumes that mainly photoreceptors are responsible for changes in retinal metabolism. Given the large photoreceptor energy consumption (*Okawa et al., 2008*; *Kanow et al., 2017*), this assumption may often be correct, yet caution is warranted to not miss metabolic responses of smaller cell populations (e.g. Müller glia, RPE cells).

## Photoreceptors may use both glucose and anaplerotic substrates as fuel

Under anaerobic conditions, the production of lactate from glucose is a hallmark of skeletal muscle function. Here, pyruvate is reduced to lactate, which enters the bloodstream to serve as substrate for liver gluconeogenesis. Glucose can then be cycled back to the muscle. This glucose-lactate cycle between muscle and liver was discovered in the 1930s and is referred to as Cori-cycle (*Cori and Cori, 1946*). The release of large amounts of lactate from isolated retina suggested that also the retina might use the Cori-cycle, albeit under aerobic conditions (*Rajala, 2020*). Photoreceptors have thus been proposed to consume primarily glucose via aerobic glycolysis (*Kanow et al., 2017*). The resultant lactate would be used as fuel in the Krebs-cycle by RPE and MGCs, generating ATP via OXPHOS. However, this hypothesis is contradicted by the high density of mitochondria in photoreceptor inner segments (*Giarmarco et al., 2020*). Moreover, key enzymes for Krebs-cycle (CS, FH, SUCLG-1) and OXPHOS (COX, ATP synthase γ) were strongly expressed in photoreceptors, but showed little or no expression in RPE or MGCs, in line with previous research (*Rueda et al., 2016*). When we cultured retina without RPE, as done by *Warburg, 1925*; *Kanow et al., 2017*, we indeed confirmed a strong lactate production. Yet, in retina cultured with RPE, lactate production was minor. Moreover, retina without RPE still produced large amounts of GTP, likely stemming from SUCLG-1 activity and indicating Krebs-cycle operation in photoreceptors. Neuroretina without RPE displayed reduced viability of both rod and cone photoreceptors, as well as an accumulation of BCAAs and lactate. In contrast, the block of glucose import into the RPE by treatment with 1,9-DDF (*Joost et al., 1988*) resulted in a depletion of BCAAs and other AAs, and an increase of retinal ATP, indicating a switch from glycolytic to anaplerotic metabolism (*Yudkoff et al., 1994*). This is in line with an earlier study using a conditional GLUT1 knockout in the RPE, which yielded evidence for a compensatory upregulation of anaplerotic metabolism in photoreceptors (*Swarup et al., 2019*). Hence, photoreceptors likely consume both glucose and anaplerotic substrates as fuels, with retinal metabolism switching to aerobic glycolysis only in the absence of RPE. This switch may be driven by increased glucose uptake.

In intact retina, tight junction-coupled RPE cells form the outer blood-retinal barrier and prevent direct access of glucose to photoreceptors (*O'Leary and Campbell, 2023*). In the absence of RPE, photoreceptors expressing the high-affinity/high-capacity GLUT3 become 'flooded' with glucose. The resultant Crabtree effect likely causes a shutdown of Krebs-cycle activity (*Simpson et al., 2008*; *Diaz-Ruiz et al., 2011*; *Crabtree, 1929*), effectively starving photoreceptors to death because a purely

glycolytic metabolism may not generate sufficient ATP for their survival. As seen in the *no* RPE group, cones appear to be more sensitive to this effect, perhaps because their overall energy demand is twice as high as that of rods (*Ingram et al., 2020*). Surprisingly, FCCP treatment did not reduce cone viability suggesting that for cones high glucose is more toxic than a lack of OXPHOS. At any rate, it is probably the lack of the blood-retinal barrier function that leads to the strong differences in metabolite patterns seen between the 1,9-DDF treatment and the *no* RPE group. Taken together, the high rates of retinal aerobic glycolysis first reported by Otto Warburg are likely an artefact of the absence of RPE, while intact retina may rely more on OXPHOS for its energy production.

## Rods need OXPHOS, cones need glycolysis

Treatment with FCCP revealed striking differences between rod and cone energy metabolism. FCCP eliminates the proton gradient between inner mitochondrial membrane and mitochondrial matrix, abolishing ATP synthesis (*Kessler et al., 1976*). Initially, FCCP may increase Krebs-cycle activity to attempt restoration of mitochondrial proton gradient (*Balcke et al., 2011*), increasing oxidative stress (*Dugan et al., 1995*), and oxidative DNA damage. The resultant activation of PARP would deplete NAD$^+$, further aggravating metabolic stress (*Schreiber et al., 2006*). Although FCCP had a strong toxic effect on rod photoreceptors, remarkably, cone photoreceptors were almost completely preserved. Theoretically, it is conceivable that FCCP primarily affected the RPE and that photoreceptor degeneration occurred secondarily to that. However, then, rods and cones would likely have been equally affected by FCCP treatment. In striking contrast to the FCCP effect was that of the 1,9-DDF block on GLUT1 (*Joost et al., 1988*). Although GLUT1 was strongly expressed in the RPE, glucose entering the RPE is shuttled forward to photoreceptors (*Swarup et al., 2019*). Remarkably, with 1,9-DDF, over 90% of cones were lost, while the detrimental effect on rods was comparatively minor. These differential effects of FCCP and 1,9-DDF treatments strongly suggest that glycolysis is sufficient and necessary for cone survival, in agreement with earlier studies who showed that under stress cones require glycolysis to remain viable (*Venkatesh et al., 2015*; *Aït-Ali et al., 2015*).

Conversely, rods require OXPHOS for their survival, while glycolysis is of minor importance. Furthermore, cone viability relative to rods was also strongly compromised in the *no* RPE situation, indicating that an intact blood-retinal barrier (*O'Leary and Campbell, 2023*) and regulation of glucose access was important for cone survival. Future studies analysing specific mouse mutants which have either only rods (e.g. the *cpfl1* mouse; *Trifunović et al., 2010*) or only cones (e.g. the NRL knockout mouse; *Mears et al., 2001*) may give more detailed insights into rod and cone metabolism.

## Expression of glucose transporters in the outer retina

Our work shows that RPE cells express high levels of GLUT1, while photoreceptors express the neuronal glucose transporter GLUT3. Furthermore, GLUT2 may contribute to photoreceptor glucose uptake indirectly via horizontal cells (*Yang et al., 2022*). We note that there is a controversy in the literature regarding the expression of GLUT1 on photoreceptors. Although there seems to be a general agreement that GLUT1 is expressed on RPE cells (*Swarup et al., 2019*), several studies have suggested GLUT1 expression also on photoreceptors, both on rods and cones (*Aït-Ali et al., 2015*; *Daniele et al., 2022*). In the brain the generally accepted setup for GLUT1 and GLUT3 expression is that low-affinity GLUT1 (Km = 6.9 mM) is expressed on glial cells, which contact blood vessels, while high-affinity GLUT3 (Km = 1.8 mM) is expressed on neurons (*Burant and Bell, 1992*; *Koepsell, 2020*). This setup matches decreasing glucose concentration with increasing transporter affinity, for an efficient transport of glucose from blood vessels to glial cells to neurons. In the retina, the cells that contact the choroidal blood vessels are the tight junction-coupled RPE cells. Since RPE cells express GLUT1, an efficient glucose transport from the RPE to the energy-hungry photoreceptors, requires photoreceptors to express a glucose transporter with a glucose affinity higher than GLUT1. Our finding of GLUT3 expression on photoreceptors matches this requirement. Still, at this point we cannot exclude a low-level expression of GLUT1 on other retinal cell types, for instance on Müller glia cells (*Actis Dato et al., 2021*) which might use GLUT1 to export excess glucose resulting from gluconeogenesis (*Goldman, 1990*) Moreover, there are 10 further glucose transporters, some of which might be expressed in the retina. A comprehensive study of the retinal protein expression of all glucose transporters, using, for example, CODEX technology (*Black et al., 2021*), may reveal their expression patterns and putative functions in the future.

## Cone photoreceptors likely employ the Cahill-cycle

The lactate-generating Cori-cycle (*Cori and Cori, 1946*) is highly inefficient and in intact retina likely plays a lesser role. An efficient pathway is the glucose-alanine cycle, or Cahill-cycle, in which pyruvate, instead of being reduced to lactate, is transaminated to alanine (*Felig, 1973*). This preserves NADH and generates α-keto acids to fuel the Krebs-cycle. The alanine generated enters the bloodstream and is taken up by the liver, where the ammonia is excreted in the form of urea, while the carbon backbone is used for gluconeogenesis. The key enzyme for the Cahill-cycle is ALT, conventionally associated with muscle and liver function (*Nathwani et al., 2005*; *Kim et al., 2008*). Previously, ALT was found in glial cells of the honeybee retina (*Balcke et al., 2011*), and ALT activity was detected in rat retinal tissue lysates (*LaNoue et al., 2001*). In our study, the localisation of ALT in photoreceptors and inner retinal neurons, combined with our interventional and metabolomic datasets, provided evidence for the operation of the Cahill-cycle in the mammalian retina. Moreover, the expression of mitochondrial PCK2 in cones facilitates the efficient uncoupling of glycolysis from the Krebs-cycle, suggesting that cones may use the Cahill-cycle for very effective energy production.

## Glutamine from MGCs may provide additional fuel for cones

MGCs are known for their uptake of extracellular glutamate and use for glutamine synthesis. In fact, in retinal histology, glutamate-aspartate transporter and GS have been widely used as markers for MGCs (*Rueda et al., 2016*; *Riepe and Norenburg, 1977*). GAC converts glutamine back to glutamate, which could then serve as a substrate for the mini-Krebs-cycle. We localised GAC in inner retinal neurons and photoreceptors, with a particularly strong expression in cone inner segments. Here, GAC overlapped with ALT expression, indicating that cones can use glutaminolysis to obtain extra glutamate for pyruvate transamination in the Cahill-cycle. This may also explain why the glycolysis inhibitor Shikonin reduced cone viability much less than that of rods. In this situation, cones might be able to use the serine-synthesis pathway – with serine production from 3-phosphoglycerate and subsequent deamination to pyruvate – to bypass the Shikonin block of PKM2.

## A hypotaurine-taurine shuttle can transfer NADH from RPE to photoreceptors

The regeneration of the photoreceptor photopigment retinal is performed by the RPE (*Redmond et al., 1998*). Recently, retinal has been proposed to form a Schiff base adduct with taurine, which would act as a retinal carrier and buffer (*Veeravalli et al., 2020*). Moreover, taurine has been suggested to be an important osmolyte in the retina that may flow in and out of neuronal cells in response to neuronal activity (*Ripps and Shen, 2012*; *Pasantes-Morales et al., 1999*).

We found that retina cultured with RPE harboured high levels of taurine and low levels of hypotaurine, while retina without RPE displayed low levels of taurine and high levels of hypotaurine. Our taurine immunostaining found essentially no taurine in the RPE, while photoreceptor inner segments and synapses displayed very high taurine levels. Together, these findings suggest a hypotaurine-taurine shuttle between RPE and photoreceptors. In the RPE taurine can be reduced to hypotaurine and shuttled back to photoreceptors where oxidation by hypotaurine dehydrogenase (*Veeravalli et al., 2020*; *Sumizu, 1962*) may reconstitute taurine, yielding additional NADH for OXPHOS. The net effect of this hypotaurine-taurine shuttle would be a transfer of reducing power from RPE to photoreceptors, boosting photoreceptor ATP production via OXPHOS. In addition, it seems likely that hypotaurine-taurine shuttling will have an impact on the osmolarity of both photoreceptors and RPE cells.

## Rods can use the mini-Krebs-cycle to uncouple glycolysis from mitochondrial metabolism

A key problem in the understanding of cellular energy metabolism is the relationship between fast but inefficient glycolysis and slow but efficient Krebs-cycle/OXPHOS (*Zheng, 2012*; *Pfeiffer et al., 2001*). Both pathways are coupled via pyruvate and the different metabolic flow rates reduce the efficiency of energy production, e.g., via feedback inhibition (*Lai and Behar, 1993*). Pyruvate coupling is especially problematic in high and rapidly changing energy demand such as in neurons and photoreceptors (*Du et al., 2016*). Uncoupling glycolysis and Krebs-cycle via the Cori-cycle is extremely wasteful and likely insufficient to satisfy long-term photoreceptor energy demand. By comparison, the Cahill-cycle

delivers additional NADH when using pyruvate derived from glycolysis, and cones may use the Cahill-cycle for uncoupling from glycolysis.

An alternative pathway is the mini-Krebs-cycle, essentially an oxalacetate to α-ketoglutarate shunt (*Yudkoff et al., 1994*). This cycle uses glutamate, glutamine, and BCAAs as fuels to run mitochondrial respiration independent of glycolysis. The key step of the mini-Krebs-cycle is the transamination of oxaloacetate/glutamate to aspartate/α-ketoglutarate by AST. α-Ketoglutarate is then metabolised to oxaloacetate, generating NADH, $FADH_2$, and GTP. Acetyl-CoA generated from BCAAs or pyruvate is used to create NAA, the end product of the mini-Krebs-cycle. We found mitochondrial AST2 to be expressed in rod inner segments, in agreement with an early cytochemical study (*Gebhard, 1991*). This indicates that the mini-Krebs-cycle is used primarily by rods, perhaps explaining their selective vulnerability to FCCP treatment. Crucially, the mini-Krebs-cycle is more energy-efficient than the Cahill-cycle and generates NAA instead of alanine as net product. Both metabolites serve the purpose of disposing of excess ammonia originating from AA input (*Moffett et al., 2007*; *Dadsetan et al., 2013*). NAA in the human brain is one of the most abundant metabolites and is routinely used in clinical MRI diagnosis to visualise brain health (*Igarashi et al., 2015*). While different functions have been hypothesised for NAA (*Yan et al., 2003*; *Moffett et al., 2013*), our work proposes NAA as a signature metabolite for the mini-Krebs-cycle. Indeed, a recent study using Raman spectroscopy imaging detected high levels of NAA in the human retina, demonstrating in vivo use of this cycle (*Alba-Arbalat et al., 2021*). Importantly, the 6-step mini-Krebs-cycle is significantly faster than the 10-step Krebs-cycle (*Yudkoff et al., 1994*), more energy-efficient, and uncoupled from glycolysis. While the high metabolic rate of photoreceptors suggests that mini-Krebs-cycle metabolites will mostly originate there, the strong AST1/AST2 expression observed in the inner plexiform layer suggests that the mini-Krebs-cycle may also play a role in fuelling synaptic activity. Moreover, in rods, PCK1 may allow to replenish cytoplasmic acetyl-CoA pools from oxaloacetate, if BCAA-derived input was insufficient.

In the future, it will be very interesting to perform carbon- or nitrogen-tracing studies using [13]C-or [15]N-labelled alternative fuels, including glucose, lactate, and glutamate/glutamine, to measure which metabolites are produced under what experimental conditions. Such tracing studies would also allow to establish the flux of metabolites through a specific pathway, something that is difficult to assess with our current steady-state measurements.

Overall, the different pathways outlined here provide photoreceptor cells with remarkable versatility and flexibility in terms of substrate use, enabling them to dynamically adapt the timing and quantitative output of energy metabolism. Rods rely strongly on the Krebs-cycle and OXPHOS, while cones are more dependent on glycolysis. These differences between rod and cone metabolism may be related to their response kinetics and sensitivities. The flexible uncoupling of glycolysis from mitochondrial respiration allows both processes to run at optimum, producing the characteristic signature metabolites lactate (Cori-cycle), alanine (Cahill-cycle), and NAA (mini-Krebs-cycle). These metabolites can reveal energy or disease status and could serve as readout and guide for the design of novel therapeutic interventions. Given the general importance of energy metabolism, the significance of our findings extends beyond the retina, for instance, to other neurodegenerative and metabolic diseases.

## Materials and methods
### Animals

C3H WT mice were used (*Sanyal and Bal, 1973*). These animal lines are regularly screened for genetic mutations known to cause retinal degeneration (e.g. *rd1, rd2, rd8, rd10, cpfl1*, etc.) and were shown to be free of such mutations. All efforts were made to minimise the number of animals used and their suffering. Animals were housed under standard white cyclic lighting, had free access to food and water, and were used irrespective of gender. Protocols compliant with the German law on animal protection were reviewed and approved by the "Einrichtung fur Tierschutz, Tierärztlichen Dienst und Labortierkunde" of the University of Tübingen (Registration Nos: AK02-19M, AK05-22M) and were following the association for research in vision and ophthalmology (ARVO) statement for the use of animals in vision research. Animals were not assigned to experimental groups prior to their sacrifice.

## Retinal explant cultures

The retinal explantation procedure and long-term cultivation in defined medium, free of serum and antibiotics, is described in detail in *Belhadj et al., 2020* (*Figure 1—figure supplement 2B*). Briefly, mice were decapitated at P9 and the heads cleaned with 70% ethanol. The eyes were removed under aseptic conditions, and placed into R16 basal medium (BM; Gibco, Paisley, UK), washed for 5 min, followed by a 15 min incubation in 0.12% proteinase K (Sigma-Aldrich, Taufkirchen, Germany; P6556) at 37°C to predigest the sclera and to allow for an easy separation of the retina together with its RPE. Then, the eyes were placed for 5 min in BM with 10% foetal calf serum (FCS) to deactivate proteinase K. In the case of no RPE explants, the retina was explanted directly, without proteinase K pre-treatment and FCS deactivation. Under the microscope and sterile conditions, the anterior segment, lens, and vitreous body were carefully removed from the eyeballs, the optic nerve was cut, and the retinas were removed from the sclera. Then, four incisions were made into the retina to give a flat, clover leaf-like structure that was transferred to a culturing membrane (sterile 24 mm insert with 0.4 µm pore size polycarbonate membrane, Corning-Costar, New York, NY, USA), with the ganglion cell layer facing up. Subsequently, culturing membranes were placed in six-well culture plates (BD Biosciences, San Jose, CA, USA) and incubated in 1 mL of R16 complete medium (CM) with supplements (*Pfeiffer et al., 2001*), at 37°C, in a humidified incubator with 5% $CO_2$. From P9 to P11 cultures were kept in CM without treatment to adapt to in vitro conditions. Drug treatments were applied to the culturing medium from P11 until P15 with either 50 µM 1,9-DDF (Sigma-Aldrich), 4 µM Shikonin (*Zhao et al., 2018*; *Chen et al., 2011*; *Traxler et al., 2022*) (Selleck Chemicals, Planegg, Germany), 20–400 µM β-chloro-alanine (Sigma-Aldrich), or 0.1–10 µM Compound 968 (Biomol, Hamburg, Germany). The FCCP treatment (5 µM) was applied from P13 to P15. This form of drug application meant that the drugs reached the RPE and neuroretina from the basal side (i.e. through the porous culturing membrane). The medium was changed every 2 days.

For all treatment conditions, culturing was stopped at P15 by 45 min fixation in 4% paraformaldehyde. Retinal tissues were cryoprotected with graded sucrose solutions containing 10%, 20%, and 30% sucrose and then embedded in Tissue-Tek O.C.T. compound (Sakura Finetek Europe, Alphen aan den Rijn, Netherlands). Tissue sections of 12 µm were prepared using Thermo Scientific NX50 microtome (Thermo Scientific, Waltham, MA, USA) and thaw-mounted onto Superfrost Plus glass slides (R. Langenbrinck, Emmendingen, Germany).

## Cell death detection (TUNEL assay)

Fixed slides were dried at 37°C for 30 min and washed in phosphate-buffered saline (PBS) solution at room temperature (RT), for 15 min. Afterwards, the slides were placed in Tris buffer with proteinase K at 37°C for 5 min to inactivate nucleases. The slides were then washed with Tris buffer (10 mM Tris-HCl, pH 7.4), three times for 5 min each. Subsequently, the slides were placed in ethanol-acetic acid mixture (70:30) at –20°C for 5 min followed by three washes in Tris buffer and incubation in blocking solution (10% normal goat serum, 1% bovine serum albumin, 1% fish gelatin in 0.1% PBS-Triton X-100) for 1 hr at RT. Lastly, the slides were placed in the TUNEL solution (labelling with either fluorescein or tetra-methyl-rhodamine; Roche Diagnostics GmbH, Mannheim, Germany) in 37°C for 1 hr and mounted with Vectashield with DAPI (Vector, Burlingame, CA, USA) thereafter.

## Immunofluorescence

Fixed slides were dried at 37°C for 30 min and rehydrated for 10 min in PBS at RT. For immunofluorescent labelling, the slides were incubated with blocking solution (10% normal goat serum, 1% bovine serum albumin in 0.3% PBS-Triton X-100) for 1 hr at RT. The primary antibodies (*Table 1*) were diluted in blocking solution and incubated at 4°C overnight. The slides were then washed with PBS, three times for 10 min each. Subsequently, the secondary antibody, diluted in PBS (*Table 1*), was applied to the slides, and incubated for 1 hr at RT. Lastly, the slides were washed with PBS and mounted with Vectashield with DAPI (Vector).

## Microscopy, cell counting, and statistical analysis

Fluorescence microscopy was performed with a Z1 Apotome microscope equipped with a Zeiss Axiocam digital camera (Zeiss, Oberkochen, Germany). Images were captured using Zen software (Zeiss) and the Z-stack function (14-bit depth, 2752*2208 pixels, pixel size = 0.227 µm, 9 Z-planes

at 1 μm steps). The raw images were converted into maximum intensity projections (MIF) using Zen software and saved as TIFF files.

Photoreceptors stained by the TUNEL assay were counted manually on three images per explant, the average cell number in a given ONL area was estimated based on DAPI staining and used to calculate the percentage of TUNEL-positive cells. Cones stained with Arr3 were counted on Apotome images using MIF. Cone numbers are expressed as Arr3-positive cells visible in 100 μm stretches of retinal circumference.

Adobe Photoshop CS6 (Adobe Systems Inc, San Jose, CA, USA) and Adobe Illustrator CC 2019 software was used for primary image processing. All data given represent the means and standard deviation from at least five different animals. Statistical comparisons between experimental groups were made using Student's paired t-test or ANOVA and multiple comparisons correction with Tukey's post hoc test (*Figure 1*) using GraphPad Prism 9.1 for Windows (GraphPad Software, La Jolla, CA, USA). Levels of significance were: $*=p<0.05$; $**=p<0.01$; $***=p<0.001$.

## Metabolite extraction

After retinal explant culture, at P15, the tissue was quickly transferred into 80% methanol/20% ethanol, snap-frozen in liquid nitrogen. A sample of the culture medium was taken from the same well plate as the retinal tissue, and snap-frozen in liquid nitrogen. Retinal tissue was placed in 400 μL of methanol (LC-MS grade), transferred to the 2 mL glass Covaris system-compatible tubes and 800 μL of methyl-tert-butyl ether (MTBE) was added, thoroughly mixed, and further subjected to metabolite extraction via ultrasonication (Covaris E220 Evolution, Woburn, MA, USA). After the extraction, 400 μL of ultra-pure water were added for two-phase liquid separation. The aqueous phase was separated and evaporated to dryness. Similarly, 400 μL of the aqueous medium sample was transferred to the 2 mL glass Covaris system-compatible tube, 800 μL of MTBE was added and subjected to the ultrasonication extraction protocol. Finally, 400 μL of methanol were added and mixed, centrifuged, and after two-phase separation the aqueous layer was separated and evaporated, to obtain a dry metabolite pellet.

## Sample preparation for ¹H-NMR spectroscopy measurements and data analysis

Dried metabolite pellets were resuspended in a deuterated phosphate buffer (pH corrected for 7.4) with 1 mM of 3-(trimethylsilyl) propionic-2,2,3,3-d₄ acid sodium salt (TSP) as internal standard. NMR spectra were recorded at 298 K on a 14.1 T ultra-shielded NMR spectrometer at 600 MHz proton frequency (Avance III HD, Bruker BioSpin, Ettlingen, Germany) equipped with a triple resonance 1.7 mm RT microprobe. Short zero-go, 1D nuclear Overhauser effect spectroscopy (NOESY), and Carr-Purcell-Meiboom-Gill (CPMG; 4096 scans for retinal tissue samples, 128 for medium samples) pulse programmes were used for spectra acquisition. Spectra were processed with TopSpin 3.6.1 software (Bruker BioSpin).

## Quantification of metabolomic data and statistical analysis

Retina tissue metabolite assignment and quantification was done on the pre-processed CPMG spectra and performed with Chenomx NMR Suite 8.5 (Chenomx Inc, Edmonton, Canada). For absolute quantification, metabolite matches were calibrated to the internal 1 mM TSP standard. Of note, not all central metabolic pathways are detectable by ¹H-NMR, which is why the fully annotated metabolite list is lacking some of the glycolytic (e.g. glycerylaldehyde-3-phosphate) and Krebs-cycle (e.g. succinyl-CoA) metabolites. After manual assignment and quantification of each sample, a raw metabolite concentration table (comma separated value) was exported and used as upload file for statistical analysis with MetaboAnalyst 5.0 platform (https://www.metaboanalyst.ca; *Pang et al., 2021*).

Highly abundant metabolites (glucose, lactate) could also be quantified in medium samples. However, in these samples the internal standards were not well preserved due to the high salt content of the medium. Therefore, spectra were manually integrated in the TopSpin software, to obtain absolute integral values for glucose (5.2 ppm) and lactate (4.1 ppm). The glucose peak integral was assigned to the known 19.1 mM absolute glucose concentration in the unspent, complete R16 medium, and glucose and lactate concentrations in experimental samples were calculated in proportion to their absolute integrals.

Raw metabolite concentrations were normalised by the probabilistic quotient normalisation method on the control group to account for dilution effects. Heatmaps display auto-scaled metabolite concentrations (red – high, blue – low). Unsupervised hierarchical cluster analysis was performed by Euclidean distance measure and Ward's distance measure. The $\log_2$ fold change (FC) shown in Volcano plots indicates the $\log_2$ (mean metabolite concentration in condition A) minus $\log_2$ (mean metabolite concentration in condition B), where A and B refer to the two different experimental groups being compared (e.g. control vs. no RPE).

Pattern hunter correlation analysis was performed on respective datasets in each specific comparison of selected metabolites based on the most statistically significant changes and importance using Pearson r distance measure for correlation coefficient calculation. GraphPad Prism 9.1.0 software (GraphPad Software, San Diego, CA, USA) was used for statistical analysis and data visualisation. For multiple group comparisons, an ordinary one-way ANOVA with Tukey's multiple comparisons post hoc test was applied. In *Figure 6—figure supplement 1*, an ordinary one-way ANOVA with Fisher's LSD test was used. For two-group comparisons, Student's unpaired, two-tailed t-test was used. FC threshold was set to >1.2, and raw p-value<0.05, p-values represent: ****<0.0001, ***<0.001, **<0.01, *<0.05. Data throughout is represented as individual data points with mean ± SD. Each mean represents at least five individual biological replicates (i.e. retinal explant cultures from different animals). Metabolite pathway analysis was based on KEGG pathway database where pathway impact values are based on certainty and pathway enrichment analysis.

## Acknowledgements

We thank N Rieger, M Owczorz, and D Bucci for first-rate technical support, as well as James B Hurley and Daniel Hass (both University of Washington, Seattle, WA, USA) for helpful comments and suggestions. This work was funded by the ProRetina Foundation, the Zinke heritage foundation, the Werner Siemens Foundation, the Chinese scholarship council (CSC), ANID-FONDECYT No. 1210790 (OS), and PhD grant BECAS CHILE/2018-21180443 (VC). We also acknowledge support from the Open Access Publication Fund of the University of Tübingen.

## Additional information

### Funding

| Funder | Grant reference number | Author |
|---|---|---|
| Fondo Nacional de Desarrollo Científico y Tecnológico | 1210790 | Oliver Schmachtenberg |
| Becas Chile/2018 | 21180443 | Victor Calbiague |
| Chinese Scholarship Council | | Yiyi Chen |
| Werner Siemens-Stiftung | | Christoph Trautwein Laimdota Zizmare |
| Pro Retina-Stiftung | | Francois Paquet-Durand |
| Tistou & Charlotte Kerstan Foundation | | Francois Paquet-Durand |

The funders had no role in study design, data collection and interpretation, or the decision to submit the work for publication.

### Author contributions

Yiyi Chen, Investigation, Visualization, Writing – original draft; Laimdota Zizmare, Investigation, Visualization, Methodology, Writing – original draft, Data curation, Formal analysis, Validation, Writing – review and editing; Victor Calbiague, Investigation, Writing – original draft; Lan Wang, Investigation, Visualization; Shirley Yu, Investigation; Fritz W Herberg, Formal analysis, Writing – review and editing; Oliver Schmachtenberg, Conceptualization, Funding acquisition, Writing – review and editing;

Francois Paquet-Durand, Conceptualization, Formal analysis, Supervision, Funding acquisition, Validation, Investigation, Visualization, Writing – original draft, Project administration, Writing – review and editing; Christoph Trautwein, Conceptualization, Supervision, Funding acquisition, Validation, Methodology, Writing – original draft, Writing – review and editing

## Author ORCIDs

Laimdota Zizmare  http://orcid.org/0000-0001-7208-3659
Victor Calbiague  http://orcid.org/0000-0002-4170-0754
Oliver Schmachtenberg  http://orcid.org/0000-0001-7854-5009
Francois Paquet-Durand  https://orcid.org/0000-0001-7355-5742
Christoph Trautwein  http://orcid.org/0000-0003-4672-6395

## Ethics

Ethics approval and consent to participateProtocols compliant with the German law on animal protection were reviewed and approved by the institution for animal welfare, veterinary service and laboratory animal science (Einrichtung für Tierschutz, Tierärztlichen Dienst und Labortierkunde) of the University of Tübingen (Registration Nos: AK02-19M, AK05-22M) and were following the association for research in vision and ophthalmology (ARVO) statement for the use of animals in vision research.

Reviewer #1 (Public review): https://doi.org/10.7554/eLife.91141.3.sa1
Reviewer #2 (Public review): https://doi.org/10.7554/eLife.91141.3.sa2
Reviewer #3 (Public review): https://doi.org/10.7554/eLife.91141.3.sa3
Author response https://doi.org/10.7554/eLife.91141.3.sa4

# Additional files

## Supplementary files

• MDAR checklist

## Data availability

Analysed datasets are included in manuscript figures and figure supplements. Raw NMR spectroscopy data has been deposited on Dryad (https://doi.org/10.5061/dryad.c2fqz61hr). Materials and tools used (e.g. chemicals, software, antibodies, equipment) are publicly available from sources stated in the Materials and methods section. This paper does not report original code.

The following dataset was generated:

| Author(s) | Year | Dataset title | Dataset URL | Database and Identifier |
| --- | --- | --- | --- | --- |
| Chen Y, Zizmare L, Calbiague V, Wang L, Yu S, Herberg FW, Schmachtenberg O, Paquet-Durand F, Trautwein C | 2024 | Retinal metabolism displays evidence for uncoupling of glycolysis and oxidative phosphorylation via Cori-, Cahill-, and mini-Krebs-cycle | http://doi.org/10.5061/dryad.c2fqz61hr | Dryad Digital Repository, 10.5061/dryad.c2fqz61hr |

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
