## [Editor Report · eLife assessment]

Chen and colleagues utilize an in situ explant model of the neural retina and retinal pigment epithelium (RPE), along with small molecule inhibition of key metabolic enzymes and targeted metabolomic analysis, to decipher key differences in metabolic pathways used by rods, cones, Muller glia, and the RPE. They conclude that rods are heavily reliant on oxidative metabolism, cones are heavily reliant on glycolysis, and multiple mechanisms exist to decouple glycolysis from oxidative metabolism in the retina. This study provides **valuable** metabolomic data and insights into the metabolic flexibility of different retinal cells. However, current evidence is still **incomplete** as several of the conclusions from the paper stand in contradiction to other published findings and the authors naturally suggests experiments that will be needed in the future to validate the hypothesized pathways and refute existing published data. Such future validation includes animal models with tissue specific knockout of the key enzymes probed in the study; inhibiting the targets of this study with more than 1 small molecule that is structurally different, and at different doses and timings; using retinal explants from matured animals; performing labeled metabolite tracing experiments; and direct assessment of mitochondrial function (via OCR) under various manipulations.

---

## [Referee Report · Reviewer #1 (Public review)]

Summary:

In the resubmitted manuscript by Chen et al. entitled, "Retinal metabolism: Evidence for uncoupling of glycolysis and oxidative phosphorylation via Cori-, Cahill-, and mini-Krebs-cycle", the authors look to provide insight on retinal metabolism and substrate utilization but using a murine explant model with various pharmacological treatments in conjunction with metabolomics. The authors conclude that photoreceptors, a specific cell within the explant, which also includes retinal pigment epithelium (RPE) and many other types of cells, are able to uncouple glycolytic and Krebs-cycle metabolism via three different pathways: (1) the mini-Krebs-cycle, fueled by glutamine and branched-chain amino acids; (2) the alanine-generating Cahill-cycle; and (3) the lactate-releasing Cori-cycle. While the authors have toned down some of their bold conclusions made in the original manuscript, they did very little in the way of providing additional well-controlled experiments, including cell-specific treatments, genetic knockouts, or stable isotope tracing to support their conclusions. Rather, the authors proceed to speculate more without additional data. The major issues raised by this reviewer were not adequately addressed. As such, the conclusions continue to be highly speculative and not well supported with evidence.

Strengths of resubmission:

The resubmission toned down some of its bold statements.

Weaknesses of resubmission:

Major weaknesses of this study persist including lack of in vivo supporting data. Also, retinal explant culture metabolomics are done in neuroretina with RPE attached, which are metabolically active and can be altered by the treatments investigated herein, further confounding the claims made regarding the neuroretina. While including the RPE in the explant model is commended, it needs to be separated from the retina prior to metabolomics to get a better sense of each tissues' metabolism. Also, melanin within RPE will hinder immunofluorescence signal, so one cannot state that RPE do not express certain enzymes based solely on immunofluorescence. Pharmacologic treatments are not cell-specific as the enzymes are expressed in numerous cells within the retina and RPE, and/or the treatments have significant off-target effects (such as shikonin). So, it is difficult to ascertain that the metabolic changes are secondary to the effects on photoreceptors alone, which the authors claim. Additionally, the explants are taken at a very early age when photoreceptors are known to still be maturing. No mention or data is presented on how these metabolic changes are altered in retinal explants after photoreceptors have fully matured. Likewise, significant assumptions are made based on a single metabolomics experiment with no stable isotope tracing to support the pathways suggested. In vivo, stable-isotope retinal metabolomics are being done and have been done, so stating this technology is beyond our field is false. Therefore, the conclusions reached in this manuscript are still not supported.

---

## [Referee Report · Reviewer #2 (Public review)]

Summary:

The authors aim to learn about retinal cell specific metabolic pathways, which could substantially improve the way retinal diseases are understood and treated. They culture ex vivo mouse retinas for 6 days with 2 - 4 days of various drug treatments targeting different metabolic pathways or by removing the RPE/choroid tissue from the neural retina. They then look at photoreceptor survival, stain for various metabolic enzymes/transporters and quantify a broad panel of metabolites. While this is an important question to address, the results are not sufficient to support the conclusions.

Strengths:

The questions the authors are exploring at extremely valuable and I commend the authors and working to learn more about retina metabolism. The different sensitivity of the cones to various drugs is interesting and may suggest key differences between rod and cones. The authors also provide a thoughtful discussion of various metabolic pathways in the context of previous publications.

Weaknesses:

As the authors point out, ex vivo culture models allow for control over multiple aspects of the environment (such as drug delivery) not available in vivo. Ex vivo cultures can provide good hints as to what pathways are available between interacting tissues. However, there are many limitations to ex vivo cultures, including shifting to a very artificial culture media condition that is extremely different than the native environment of the retina. It is well appreciated that cells have flexible metabolism and will adapt to conditions provided. Therefore, observations of metabolic responses obtained under culture conditions need to be interpreted with caution, they indicate what the tissue is doing under those specific conditions (which include cells adapting and dying).

Chen et al use pharmacological interventions are to the impact of various metabolic pathways on photoreceptor survival and "long term" metabolic changes. The dose and timing of these drug treatments are not examined though. It is also hard to know how these drugs penetrate the tissue and it needs to be validated that they intended targets are being accurately hit. These relatively long term treatments should be causing numerous downstream changes to metabolism, cell function and survival, which makes looking at a snap shot of metabolite levels hard to interpret. It would be more valuable to look at multiple time points after drug treatment, especially easy time points (closer to 1 hr). the authors use metabolite ratios to make conclusions about pathway activity. It would be more valuable to directly measure pathway activity by looking a metabolite production rates in the media and/or with metabolic tracers again in time scales closer to minutes and hours instead of days.

While the data is interesting and may give insights into some rod and cone specific metabolic susceptibility, more work is needed to validate these conclusions. Given the limitations of the model the authors have over interpreted their findings and the conclusions are not supported by the results. They need to either dramatically limit the scope of their conclusions or validate these hypotheses with additional models and tools.

---

## [Referee Report · Reviewer #3 (Public review)]

Summary:

The neural retina is one of the most energetically active tissues in the body and research into retinal metabolism has a rich history. Prevailing dogma in the field is that the photoreceptors of the neural retina (rods and cones) are heavily reliant on glycolysis, and as oxygen tension at the level of photoreceptors is very low, these specialized sensory neurons carry out aerobic glycolysis, akin to the Warburg effect in cancer cells. It has been found that this unique metabolism changes in many retinal diseases, and targeting disease-altered retinal metabolism may be a viable treatment strategy. The neural retina is composed of 11 different cell types, and many research groups over the past century have contributed to our current understanding of cell-specific metabolism of retinal cells. More recently, it has been shown in mouse models and co-culture of the mouse neural retina with human RPE cultures that photoreceptors are reliant on the underlying retinal pigment epithelium for supplying nutrients. Chen and colleagues add to this body of work by studying an ex vivo culture of the developing mouse retina that maintained contact with the retinal pigment epithelium. They exposed such ex vivo cultures to small molecule inhibitors of specific metabolic pathways, performing targeted metabolomics on the tissue and staining tissue with key metabolic enzymes to lay the groundwork for what metabolic pathways may be active in particular cell types of the retina. The authors conclude that rod and cone photoreceptors are reliant on different metabolic pathways to maintain their cell viability - in particular, that rods rely on oxidative phosphorylation and cones rely on glycolysis. Further, their data suggest multiple mechanisms whereby glycolysis may occur simultaneously with anapleurosis to provide abundant energy to photoreceptors. The data from metabolomics revealed several novel findings in retinal metabolism, including the use of glutamine to fuel the mini-Krebs cycle, the utilization of the Cahill cycle in photoreceptors, and a taurine/hypotaurine shuttle between the underlying retinal pigment epithelium and photoreceptors to transfer reducing equivalents from the RPE to photoreceptors. In addition, this study provides quantitative metabolomics datasets that can be compared across experiments and groups. The use of this platform will allow for rapid testing of novel hypotheses regarding the metabolic ecosystem in the neural retina.

Strengths:

The data on differences in susceptibility of rods and cones to mitochondrial dysfunction versus glycolysis provides novel hypothesis-generating conjectures that can be tested in animal models. The multiple mechanisms that allow anapleurosis and glycolysis to run side-by-side add significant novelty to the field of retinal metabolism, setting the stage for further testing of these hypotheses as well.

Weaknesses:

Almost all of the conclusions from the paper are preliminary, based on data showing enzymes necessary for a metabolic process are present and the metabolites for that process are also present. However, to truly prove whether these processes are happening (rather than speculation of the possibility they are happening), further experiments are necessary. As it currently stands, results from this study contradict results from other studies - in particular that cones, not rods, are most reliant of glycolysis. The authors attempt to address these contradictions, but without further experimentation, logical arguments carry only so much weight. At a minimum, the authors have argued that the small molecules they use are exquisitely specific for their intended targets, but validating results with a second small molecule that hits the same target but is structurally different would bolster their claims. Genetically knocking down the intended targets with interfering RNA technology would also be possible, as would explant cultures from knock-out animals. Without these studies to confirm target specificity, combined with the fact that conclusions from this study contradict existing studies in the literature, the results have to be categorized as speculative and hypothesis-generating rather than conclusive.

---

## [Author Response]

The following is the authors’ response to the original reviews.

**Reviewer #1 (Public Review):**
Summary:In the manuscript by Chen et al. entitled, "The retina uncouples glycolysis and oxidative phosphorylation via Cori-, Cahill-, and mini-Krebs-cycle", the authors look to provide insight on retinal metabolism and substrate utilization by using a murine explant model with various pharmacological treatments in conjunction with metabolomics. The authors conclude that photoreceptors, a specific cell within the explant, which also includes retinal pigment epithelium (RPE) and many other types of cells, are able to uncouple glycolytic and Krebs-cycle metabolism via three different pathways: (1) the mini-Krebs-cycle, fueled by glutamine and branched-chain amino acids; (2) the alanine-generating Cahill-cycle; and (3) the lactate-releasing Cori-cycle. While intriguing if determined to be true, these cell-specific conclusions are called into question due to the ex vivo experimental setup with the inclusion of RPE, the fact that the treatments were not cell-specific nor targeted at an enzyme specific to a certain cell within the retina, and no stable isotope tracing nor mitochondrial function assays were performed. Hence, without significant cell-specific methods and future experimentation, the primary claims are not supported.Strengths:This study attempts to improve on the issues that have limited the results obtained from previous ex vivo retinal explant studies by culturing in the presence of the RPE, which is a major player in the outer retinal metabolic microenvironment. Additionally, the study utilizes multiple pharmacologic methods to define retinal metabolism and substrate utilization.Weaknesses:A major weakness of this study is the lack of in vivo supporting data. Explant cultures remove the retina from its dual blood supply. Typically, retinal explant cultures are done without RPE. However, the authors included RPE in the majority of experimental conditions herein. However, it is unclear if the metabolomics samples included the RPE or not. The inclusion of the RPE, which is metabolically active and can be altered by the treatments investigated herein, further confounds the claims made regarding the neuroretina. Considering the pharmacologic treatments utilized with the explant cultures are not cell-specific and/or have significant off-target effects, it is difficult to ascertain that the metabolic changes are secondary to the effects on photoreceptors alone, which the authors claim. Additionally, the explants are taken at a very early age when photoreceptors are known to still be maturing. No mention or data is presented on how these metabolic changes are altered in retinal explants after photoreceptors have fully matured. Likewise, significant assumptions are made based on a single metabolomics experiment with no stable isotope tracing to support the pathways suggested. While the authors use immunofluorescence to support their claims at multiple points, demonstrating the presence of certain enzymes in the photoreceptors, many of these enzymes are present throughout the retina and likely the RPE. Finally, the claims presented here are in direction contradiction to recent in vivo studies that used cell-specific methods when examining retinal metabolism. No discussion of this difference in results is attempted.Response: We agree with the reviewer that in vivo studies could be very interesting indeed. However, technologically it will be extremely difficult to (repeatedly/continuously) sample the retina of an experimental animal and to combine this with an interventional study, with a subsequent metabolomic analysis. We do not currently have access to such technology nor are we aware of any other lab in the world capable of doing such studies. Moreover, virtually all prior studies on retinal metabolism have been done on explanted retina without RPE. This includes the seminal studies by Otto Warburg in the 1920s. As opposed to this, our retinal samples for also all the metabolomic analyses included the RPE, except for the no RPE condition that was used as a comparator for the earlier investigations.

We note that our metabolomic analysis was done for all five experimental conditions where each condition included at least five independent samples (each derived from different animals).

The reviewer is correct to say that our organotypic explant cultures are early post-natal, with explantation performed at post-natal day 9 and culturing until day 15. Since our retinal explant system has been validated extremely well over more than three decades of pertinent research (see for instance: Caffe et al., Curr Eye Res. 8:1083-92, 1989), we are confident that photoreceptors mature in vitro in ways that are very similar to the in vivo situation. As far as studies in adult retina (i.e. three months or older) are concerned, this is indeed an important question that will be addressed in future studies. Studies employing stable isotope labelling may also be very informative and are planned for the future, also in order to properly determine fluxes. This will likely require an extension to our NMR hardware with an 15N channel probe, something that we plan on implementing in the future.

We are aware that a number of questions relating to retinal metabolism are controversial and that the use of other methodology or experimental systems may lead to alternative interpretations. We have now included citations of other studies that use, for example, conditional and/or inducible knock-outs or in vivo blood sampling (e.g. Wang et al., IOVS 38:48-55, 1997; Yu et al., Invest Ophthalmol Vis Sci. 46:4728-33, 2005; Swarup et al., Am J Physiol Cell Physiol. 316:C121-C133, 2019; Daniele et al., FASEB Journal 36:e22428, 2022) and discuss the pros and cons of such approaches (e.g. in Lines 376-384; 454-472).

**Reviewer #2 (Public Review):**
Summary:The authors aim to learn about retinal cell-specific metabolic pathways, which could substantially improve the way retinal diseases are understood and treated. They culture ex vivo mouse retinas for 6 days with 2 - 4 days of various drug treatments targeting different metabolic pathways or by removing the RPE/choroid tissue from the neural retina. They then look at photoreceptor survival, stain for various metabolic enzymes, and quantify a broad panel of metabolites. While this is an important question to address, the results are not sufficient to support the conclusions.Strengths:The questions the authors are exploring at extremely valuable and I commend the authors and working to learn more about retina metabolism. The different sensitivity of the cones to various drugs is interesting and may suggest key differences between rods and cones. The authors also provide a thoughtful discussion of various metabolic pathways in the context of previous publications.Weaknesses:As the authors point out, ex vivo culture models allow for control over multiple aspects of the environment (such as drug delivery) not available in vivo. Ex vivo cultures can provide good hints as to what pathways are available between interacting tissues. However, there are many limitations to ex vivo cultures, including shifting to a very artificial culture media condition that is extremely different than the native environment of the retina. It is well appreciated that cells have flexible metabolism and will adapt to the conditions provided. Therefore, observations of metabolic responses obtained under culture conditions need to be interpreted with caution, they indicate what the tissue is doing under those specific conditions (which include cells adapting and dying).Chen et al use pharmacological interventions to the impact of various metabolic pathways on photoreceptor survival and "long term" metabolic changes. The dose and timing of these drug treatments are not examined though. It is also hard to know how these drugs penetrate the tissue and it needs to be validated that the intended targets are being accurately hit. These relatively long-term treatments should be causing numerous downstream changes to metabolism, cell function, and survival, which makes looking at a snapshot of metabolite levels hard to interpret. It would be more valuable to look at multiple time points after drug treatment, especially easy time points (closer to 1 hr). The authors use metabolite ratios to make conclusions about pathway activity. It would be more valuable to directly measure pathway activity by looking a metabolite production rates in the media and/or with metabolic tracers again in time scales closer to minutes and hours instead of days.It is not clear from the text if the ex vivo samples with RPE/choroid intact are analyzed for metabolomics with the RPE/choroid still intact or if this is removed. If it is not removed, the comparison to the retina without RPE/choroid needs to be re-interpreted for the contribution of metabolites from the added tissue. The composition of the tissue is different and cannot be disentangled from the changes to the neural retina specifically.While the data is interesting and may give insights into some rod and cone-specific metabolic susceptibility, more work is needed to validate these conclusions. Given the limitations of the model the authors have over-interpreted their findings and the conclusions are not supported by the results. They need to either dramatically limit the scope of their conclusions or validate these hypotheses with additional models and tools.

Response: We thank the reviewer for the insightful comments and agree that some of our interpretations may have been phrased too determinedly. We have therefore rephrased and toned down our conclusions in many instances in the text, and changed the manuscript title to now read “Retinal metabolism: Evidence for uncoupling of glycolysis and oxidative phosphorylation via Cori-, Cahill-, and mini-Krebs-cycle”.

Nevertheless, when considering the major known metabolic pathways and their possible impact on metabolite patterns after the experimental manipulations used here, we believe our interpretations to be consistent with the data obtained. Conversely, the previously suggested retinal aerobic glycolysis cannot explain most of the data we have obtained. Even further, also a predominant use of the classical “full” Krebs-cycle/OXPHOS would not explain the metabolite patterns found (e.g. alanine, N-acetylaspartate (NAA)). While this does not in itself mean that our interpretations are all correct, they seem plausible in view of the data at hand and will hopefully stimulate further research on retinal energy metabolism using complementary technologies that were not available to us for the purpose of this study.

We comment that our organotypic retinal explant cultures, while they do contain their very own, native RPE, do not comprise the choroidal vasculature (in our explantation procedure the RPE readily detaches from the choroid).

As far as the drugs used on retinal explants are concerned, we note that:

(1) all three compounds used are extremely well validated, with literally thousands of studies and decades of research to their credit (i.e., 1,9-dideoxyforskolin: >270 publications since 1984; Shikonin: >1000 publications since 1977; FCCP: >2800 publications since 1967),

(2) all experimental conditions show clear and differential drug effects, as shown, for instance, by the principal component analysis in Figure1I and the cluster analysis in Figure2A,

(3) the response patterns observed for key metabolites match the anticipated drug effects (e.g. decreased glucose consumption with 1,9-dideoxyforskolin; decreased lactate levels with Shikonin; lactate accumulation with FCCP).

One can therefore be reasonably certain that these drugs did penetrate the explanted retina and that their respective drug targets were hit. Assessing dose-responses would certainly be interesting, however, the aim of this initial study was not pharmacodynamics but a general manipulation of energy metabolism. Moreover, given the extensive validation of these drugs, off-target effects seem not very likely at the concentrations used.

We agree with the reviewer that using a longitudinal, time-series type of analysis could give additional insights. We note that each additional time-point will require retinae from 25 animals and a very resource-intensive and time-consuming metabolomic analysis, together with a significantly more complex multivariate analysis (metabolite, experimental condition, time). This is a completely new undertaking that is simply not feasible as an extension of the present study.

To look at pathway activity in more direct ways is very good idea, to this end we aim to implement in the future an idea put forward by the reviewers, namely 13C-labeling and additionally 15N-labeling and tracing for specific metabolic fuels (e.g. glucose, lactate and anaplerotic amino acids such as glutamate and branched chain amino acids).

The reviewer is of course correct to say that the culture condition is somewhat artificial and that this may have introduced changes in the metabolism. However, as noted above in the first response to reviewer #1, the organotypic retinal culture system, using a defined medium, free of serum and antibiotics, has been extremely well studied and validated for decades (cf. Caffé et al., Curr Eye Res. 8:1083-92, 1989). Importantly, this system allows to maintain retinal viability, histotypic organization, and function over many weeks in culture. Moreover, most previous studies on retinal metabolism have also used explanted retina – acute or cultured – i.e. experimental approaches that are similar to what we have used and that may be liable to their own artefactual changes in metabolism. This includes the seminal, 1920s studies by Otto Warburg, or the 1980s studies by Barry Winkler, the results of which the reviewers do not seem to doubt.

We further agree that studying retinal metabolism in a situation closer to in vivo conditions would be thrilling, however to our knowledge to date there is no retina model that fully mimics the complex interplay of the blood metabolome with metabolic tissue activity. This likely means that for each metabolic condition to study (e.g. hyperglycemia, cachexia, etc.), a fairly large number of animals will need to be sacrificed for the molecular investigation of ex vivo retinal biopsies, which would mean a tremendous animal burden.

We hope the reviewer will appreciate that the revised manuscript now includes numerous improvements, along with new, additional datasets and figures, references to further relevant literature, and – as mentioned above – a more cautious phrasing of our interpretations and conclusions, including a more careful wording for the manuscript title.

**Reviewer #3 (Public Review):**
Summary:The neural retina is one of the most energetically active tissues in the body and research into retinal metabolism has a rich history. Prevailing dogma in the field is that the photoreceptors of the neural retina (rods and cones) are heavily reliant on glycolysis, and as oxygen tension at the level of photoreceptors is very low, these specialized sensory neurons carry out aerobic glycolysis, akin to the Warburg effect in cancer cells. It has been found that this unique metabolism changes in many retinal diseases, and targeting retinal metabolism may be a viable treatment strategy. The neural retina is composed of 11 different cell types, and many research groups over the past century have contributed to our current understanding of cell-specific metabolism of retinal cells. More recently, it has been shown in mouse models and co-culture of the mouse neural retina with human RPE cultures that photoreceptors are reliant on the underlying retinal pigment epithelium for supplying nutrients. Chen and colleagues add to this body of work by studying an ex vivo culture of the developing mouse retina that maintained contact with the retinal pigment epithelium. They exposed such ex vivo cultures to small molecule inhibitors of specific metabolic pathways, performing targeted metabolomics on the tissue and staining the tissue with key metabolic enzymes to lay the groundwork for what metabolic pathways may be active in particular cell types of the retina. The authors conclude that rod and cone photoreceptors are reliant on different metabolic pathways to maintain their cell viability - in particular, that rods rely on oxidative phosphorylation and cones rely on glycolysis. Further, their data support multiple mechanisms whereby glycolysis may occur simultaneously with anapleurosis to provide abundant energy to photoreceptors. The data from metabolomics revealed several novel findings in retinal metabolism, including the use of glutamine to fuel the mini-Krebs cycle, the utilization of the Cahill cycle in photoreceptors, and a taurine/hypotaurine shuttle between the underlying retinal pigment epithelium and photoreceptors to transfer reducing equivalents from the RPE to photoreceptors. In addition, this study provides robust quantitative metabolomics datasets that can be compared across experiments and groups. The use of this platform will allow for rapid testing of novel hypotheses regarding the metabolic ecosystem in the neural retina.Strengths:The data on differences in the susceptibility of rods and cones to mitochondrial dysfunction versus glycolysis provides novel hypothesis-generating conjectures that can be tested in animal models. The multiple mechanisms that allow anapleurosis and glycolysis to run side-by-side add significant novelty to the field of retinal metabolism, setting the stage for further testing of these hypotheses as well.Weaknesses:Almost all of the conclusions from the paper are preliminary, based on data showing enzymes necessary for a metabolic process are present and the metabolites for that process are also present. However, to truly prove whether these processes are happening, C13 labeling or knock-out or over-expression experiments are necessary. Further, while there is good data that RPE cultures in vitro strongly recapitulate RPE phenotypes in vivo, ex vivo neural retina cultures undergo rapid death. Thus, conclusions about metabolism from explants should either be well correlated with existing literature or lead to targeted in vivo studies. This paper currently lacks both.

Response: As mentioned above in the first answers to reviewers #1 and #2, we think of our study as a starting point that may provide novel directions for a whole series of investigations into retinal energy metabolism. Especially the use of novel technologies may in the future allow to decipher the different metabolic phenotypes of the 100+ distinct retinal cell types by in situ spatial metabolomics and lipidomics. Currently, we still have to limit the scope of our studies to only certain aspects of this topic. We thus agree that some of our interpretations need to be formulated more carefully and we have done so in the revised version of our manuscript. We also agree with the reviewer that carbon (13C) labelling and tracing studies will be very informative and will engage in such studies in the future. Besides 13C, we aim to further employ 15N labelled substrates, which is especially suitable to study the destiny of amino acids.

As far as our organotypic retinal explant system is concerned, it is arguably one of the best validated such systems available (see responses to reviewers #1 and #2). While the reviewer is correct to say that the neuroretina without RPE degenerates relatively quickly in vitro, in our system, with the neuroretina and its native RPE cultured together, we can routinely culture the retina for four weeks or more, without major cell loss (Söderpalm et al., IOVS 35:3910-21, 1994; Belhadj et al., JoVE 165, 2020). Thus, our retinal cultures with RPE do not undergo rapid death. Within the time-frame of the present study (6 days in vitro) culturing-induced cell death is minimal and unlikely to influence our analyses. For further, more detailed answers to the reviewers’ questions please see our detailed point-to-point response below.

We agree with the reviewer that eventually in vivo studies will be important to confirm our interpretations. As mentioned in our initial response to reviewer #1, such studies will be very challenging and new technologies may need to be developed before in vivo investigations can deliver the answers to the questions at hand (see answer to question Rev#3.17 below), especially if the cross-play between substrate availability from the blood metabolome and the retinal metabolic pathway activity shall be studied.

**Recommendations For The Authors**

**Reviewer #1 (Recommendations For The Authors):**
Rev#1.1. The animals should be screened for and lack rd8.

Response: This is a pertinent question from the reviewer. Ever since we first became aware of the presence of rd8 mutations in certain mouse lines from major vendors (e.g. Charles River, Jackson Labs) in around 2010, we have setup regular screening of all our mouse lines for this Crb1 mutation. Accordingly, the mouse lines used in this study were confirmed to be free of the rd8 / Crb1 mutation. A corresponding remark has now been inserted into the SI materials and methods section (Lines 37-38).

Rev#1.2. GLUT1 looks significantly different from in vivo to in vitro. Recommend co-staining with RHO and cone markers (PNA or CAR) to further delineate where it is being expressed. The in vitro cultures appear to have much shorter outer segments (OS). Considering OS biosynthesis is thought to drive a good deal of metabolic adaptations, how relevant is the in vitro model system to what is truly occurring in vivo?

Response: The GLUT1 staining shown in Figure 1 displays the in vivo situation. Since may not have been entirely clear from the previous figure legend, we have now labelled this as “in vivo retina” and distinguish it from “in vitro” samples in the legend to Figure 1 (Lines 774-778). As far as the comparison of GLUT1 staining in vivo (Figure 1A3) vs. in vitro (Figure S1C3) is concerned, in both situations a strong RPE labelling is clearly visible, with essentially no GLUT1 label within the neuroretina.

Nevertheless, to better delineate the expression of GLUT1 in the outer retina, we have now performed an additional co-staining with rhodopsin (RHO) as rod marker and peanut agglutinin (PNA) as cone marker, as suggested by the reviewer (new supplemental Figure S1). In brief, this co-staining confirms the strong expression of GLUT1 in the RPE, while there is essentially no GLUT1 detectable in rod or cone photoreceptors.

Retinal explants in long-term cultures do indeed have somewhat shorter outer segments compared to same age in vivo counterparts (Caffe et al., Curr Eye Res. 8:1083-1092, 1989). However, in the short-term cultures (6 DIV) and at the age studied here (P15) outer segments have only just started to grow out and are around 10 - 12 µm long, both in vitro and in vivo (cf. LaVail, JCB 58:650-661, 1978). Thus, the metabolism required for outer segment synthesis should be equivalent when in vitro and in vivo situations are compared. For considerations on outer segments in retinal explant cultures see also Rev#3.2 and Rev#3.29.

Rev#1.3. Also, recent publications have shown that GLUT1 is expressed in the neuroretina including rods, cones, and muller glia. Was GLUT1 not appreciated in these cells in your ex vivo samples and if so, why? Likewise, these same studies previously demonstrated GLUT1 resulted in rod degeneration but not cone. The results presented here differ significantly. Why the difference in results and is it secondary to the in vitro vs. in vivo setting? Furthermore, the authors state that they thought the no RPE situation would be similar to the GLUT1 inhibitor experimental condition but instead, they were vastly different. Is this secondary to the fact that GLUT1 is expressed outside the RPE.

Response: We are aware that there is a controversy regarding GLUT1 expression in the neuroretina, please see also our response to question Rev#3.1 below. As far as our immunostaining for GLUT1 on in vivo retina is concerned, we find an unambiguous and very marked expression of GLUT1 in RPE cells, at both basal and apical sides. Compared to the RPE, the neuroretina appears devoid of GLUT1 staining. However, at very high gamma values a faint staining in the neuroretina becomes visible, a staining which from its appearance – processes spanning the entire width of the retina – is most compatible with Müller glia cells. Under normal circumstances we would have dismissed such a faint staining as background and false positive. Given the sometimes very contradicting reports in the literature, we cannot fully exclude a weak expression of GLUT1 also in cells other than the RPE, with Müller glial cells perhaps being the most likely candidate. At any rate, GLUT1 expression in the neuroretina can only be much weaker than in the RPE, making its relevance for overall retinal metabolism unclear.

As far as recent publications studying GLUT1 in the retina are concerned, we know of the study by Daniele et al. (FASEB Journal 36:e22428, 2022), which used a rod-specific, conditional knock-out of GLUT1 and found a relatively slow rod degeneration. We are not aware of a selective GLUT1 knock-out in cones, nor are we aware of conditional GLUT3 knock-outs in the retina. For further discussion of the Daniele et al. study please see Rev#3.13.

The reviewer is right, initially we were thinking that, since GLUT1 was expressed only (predominantly) in RPE, the metabolic response to GLUT1 inhibition should look similar to the no RPE situation. However, this initial hypothesis did not consider a key fact: The RPE builds the blood retinal barrier and the tight-junction coupled RPE cells are a barrier to any larger molecule, including glucose. Removing the barrier by removing the RPE dramatically increases the availability of glucose to the retina, a phenomenon that is likely exacerbated by the expression of the high affinity/high capacity GLUT3 on photoreceptors (cf. Figure S1A). In other words, when the RPE is removed the outer retina is “flooded” with glucose and we believe that this is probably the main factor that explains why the metabolic response to GLUT1 inhibition (1,9-DDF group) is so different from the no RPE condition.

We have now included an additional corresponding explanation in the discussion (Lines 422-429). Furthermore, we have added an entire new subchapter to the discussion to debate the expression of glucose transporters in the outer retina (Lines 454-472).

Rev#1.4. Shikonin's mechanism of action via protein aggregation and lack of specificity for PKM2 vs PKM1 at 4uM is an experimental limitation that needs to be taken into account. All treatments utilized are not cell-specific.

Response: While the reviewer is correct to say that Shikonin may have multiple cellular targets and a diverse range of possible applications as an anti-inflammatory, antimicrobial, or anticancer agent (cf. Guo et al., Pharmacol. Res. 149:104463, 2019), numerous studies support its specificity for PKM2 over PKM1, at concentrations ranging from 1 – 10 µM (Chen et al., Oncogene 30:4297-306, 2011; Zhao et al., Sci. Rep. 8:14517, 2018; Traxler et al., Cell Metab. 34:1248-1263, 2022). We settled for 4 μM as an intermediate concentration, considering its effectiveness and specificity in previous studies. We have now inserted references detailing the specificity and concentration range of Shikonin into the SI Materials and Methods section (Line 62).

The concern that “all treatments” are not cell-specific is debatable. Certainly, any given compound may have off-target effects, yet, since the compounds we used in our study have all been studied for decades (see above, initial response to Reviewer #2), their off-target profile is well established and unlikely to play an important role here. Moreover, in our study the cell specificity does not come from the compounds used but from where their targets are expressed. As shown in Figure 1A and in Figure S1C, Shikonin´s target PKM2 is almost exclusively expressed in photoreceptor inner segments. Hence, it seems very reasonable to expect that the vast majority of the metabolomic changes observed by Shikonin treatment are related to photoreceptors. We note that this assertion would still be true even if there was a low-level expression of PKM2 in other retinal cell types and/or if Shikonin had moderate off-target effects on other enzymes since the bulk of the effect on the quantitative metabolomic dataset would still originate from PKM2 inhibition in photoreceptors.

Rev#1.5. What was the method of cone counting in Figure 1?

Response: Cones were counted per 100 µm of retinal circumference based on an arrestin-3 staining (cone arrestin, CAR).

This information is now included in the SI Materials and Methods section under “Microscopy, cell counting, and statistical analysis” (Lines 99-100).

Rev#1.6. How do you know that FCCP is not altering RPE ox phos, disrupting the outer retinal microenvironment and leading to cell death, and therefore, the effects seen are not photoreceptor-specific but rather downstream from the initial insult in RPE?

Response: We propose that FCCP will be acting on both photoreceptors and RPE cells (and all other retinal cell types) at essentially the same time, over the experimental time-frame. Thus, OXPHOS should be inhibited in all cells simultaneously. However, FCCP will primarily affect cells that actually use OXPHOS to a large extent, while cells relying on other metabolic pathways (e.g. glycolysis) will hardly be affected.

We believe the very strong effect of FCCP, seen exclusively in rod photoreceptors, to be a direct drug effect. While we cannot not fully exclude an indirect effect via the RPE – as proposed by the reviewer – we think this to be unlikely because:

(1) RPE viability was not compromised by FCCP treatment.

(2) If the reviewer´s hypothesis was correct, then also cone photoreceptors should have been affected (e.g. because now the RPE consumes all glucose, leaving nothing for cones). However, cones were essentially unaffected by the FCCP treatment, making a dependence on RPE OXPHOS unlikely. Especially so, because blocking GLUT1 and glucose import on the RPE with 1,9-DDF had only relatively minor effects on rod photoreceptor viability but strongly affected cones. This indicates that the RPE is mainly shuttling glucose through to photoreceptors, especially to cones, and this function does not seem to be impaired by FCCP treatment.

(3) We found that enzymes required for Krebs-cycle and OXPHOS activity (i.e. citrate synthase, fumarase, ATP synthase γ) are predominantly expressed in photoreceptors but virtually absent from RPE (Figure 3D, see also answer to following question).

(4) The density of mitochondria (i.e. the target for FCCP) is far lower in RPE than in photoreceptors, as evidenced also by the COX staining shown in Figure 1A. Hence, photoreceptors are far more likely to be hit by FCCP treatment than RPE cells.

To accommodate the reviewer´s concern, we have now added a further comment into the discussion (Lines 440-442).

Rev#1.7. While Figure 3D is interesting, it offers no significant insight into mechanisms as the enzyme levels are not being compared to control nor is mitochondrial fitness in these conditions being assessed, which would provide greater insight than just showing that these enzymes are present in the inner segments, which are known to be rich in mitochondria. Additionally, stating that the low ATP is secondary to decreased Krebs cycle activity and ox phos based on merely ATP levels is not supported by metabolite levels minus citrate nor ox phos enzyme levels or oxygen consumption. Also, citrate is purported to be decreased in the table in Figure 2 in the no RPE condition; however, Supplemental Figure 2 demonstrates this change is not significant then the same data is presented in Supplemental Figure 3 and it is statistically significant again. Why the difference in data and why is the same data being shown multiple times?

Response: The immunostaining shown in Figure 3D shows the in vivo retina, or in other words the localization of enzymes in the native situation. Since this may not have been obvious in the previous manuscript version, we have added a corresponding comment to the legend of Figure 3 (Line 806). The localization of the Krebs-cycle/OXPHOS enzymes citrate synthase, fumarase, and ATP synthase mainly to photoreceptors, but not (or much less) to RPE, is another piece of evidence supporting the idea that OXPHOS is predominantly performed by photoreceptors (see also answer to previous question Rev#1.6).

The decreased ATP levels (together with citrate, aspartate, NAA) shown in Figure 3 in the no RPE group, are an indication that photoreceptor Krebs-cycle activity may be decreased but not abolished in the absence of RPE. Importantly, GTP levels are not reduced in the no RPE group (Figure 2). Since large amounts of GTP can only by synthesized by either SUCLG-1 in the Krebs-cycle or by NDK-mediated exchange with ATP, the most plausible interpretation is that Krebs-cycle dependent ATP-synthesis was decreased in the no RPE situation, but that the (mini) Krebs-cycle or Cahill-cycle, notably the step from succinyl-CoA to succinate, was running. Since there is no RPE in this group, this strongly suggests important Krebs-cycle/OXPHOS activity in photoreceptors where the majority of the corresponding enzymes are located (see above).

We thank the reviewer for pointing out that the information on group comparisons may not have been presented with sufficient clarity. In the figures mentioned by the reviewer the data is shown and compared in different contexts: the table in Figure 2B and the data in Figure S3 (now renumbered to Figure S5) refer to two-way comparisons of treatment condition to control, to elucidate individual treatment effects. Meanwhile Figure S2 (now supplementary Figure S3) refers to a 5-way comparison for a general overview that puts all five groups in context with each other. These differences in comparisons and normalization to the respective common standards entail the use of different statistical tools, resulting in different p-values. The statistical testing approaches and thresholds are now disclosed in the figure legends, and additionally in the SI Materials and Methods section (Lines 145-155).

Rev#1.8. When were the ex vivo samples taken for metabolomics, and if taken when significant TUNEL staining and cell death have occurred, are the changes in metabolism due to cell death or a true indication of differential metabolism? Furthermore, it is unclear if the metabolomics samples included the RPE or not. Considering these treatments will affect most cells in the retina and the RPE, which is included in the ex vivo samples, it is difficult to ascertain that these changes are secondary to the effects on photoreceptors alone.

Response: The samples for metabolomics included the RPE (except for the no RPE condition) and were taken at the same time as the tissues for histological preparations and TUNEL assays, i.e. they were all taken at post-natal day 15. This has now been clarified in the SI Materials and Methods section (Lines 108-110).

We cannot entirely exclude an effect of ongoing cell death caused by the different drug treatments on the retinal metabolome. However, since in the experimental treatments cell death was still comparatively low (even in the FCCP condition, overall cell death was only around 10% of the total retina), and the metabolomic analysis considered the entire tissue, the impact of cell death per se on the total metabolome will be comparatively minor (≤ 10%, i.e. within the typical error margin of the metabolomic analysis).

As mentioned above, the drug treatments should in principle affect all retinal cells at the same time. However, only cells that express the drug targets (i.e. 1,9-DDF targets GLUT1 in RPE cells, Shikonin targets PKM2 in photoreceptors; cf. Figure 1A) should react to the treatment. Even FCCP, in the paradigm employed, will only affect those cells that rely heavily on OXPHOS. Our data indicates that while this is almost certainly the case for rods; cones, RPE cells, and essentially all of the inner retina, are not affected by FCCP treatment, strongly suggesting that OXPHOS is of minor importance for these cell populations.

Rev#1.9. Why were the FCCP and no RPE groups compared? If they have similar metabolite patterns as noted in Figure 2, would that suggest that FCCP's greatest effect is on the ox phos of RPE and the metabolite patterns are secondary to alterations in RPE metabolism? Also, the increase in citrate and decrease in NAD may be related to effects on RPE mitochondrial metabolism when comparing these groups, and the disruption of RPE metabolism may then result in PARP staining of photoreceptors.

Response: The reason for the pair-wise comparison of the no RPE and FCCP groups initially was indeed the similarity in metabolite patterns. This was now rephrased accordingly in the results section “Photoreceptors use the Krebs-cycle to produce GTP” (Lines 218-219). The interpretation that the reviewer proposed here is interesting, but does not conform with the data analysis of this and other group comparisons.

Instead, the similarity between the metabolic patterns found in the no RPE and FCCP groups further supports the idea that a lack of RPE decreases retinal OXPHOS and increases glycolysis. This interpretation is based on the following observations:

(1) Mitochondrial density in the RPE is far lower than in photoreceptors (see COX staining in Figure 1A), thus quantitatively the metabolite pattern caused by a disruption of OXPHOS (via FCCP treatment) will be dominated by metabolites generated by photoreceptors. For the same reason the depletion of retinal NAD+, and the concomitant increase in photoreceptor PAR accumulation after FCCP treatment, is unlikely to be due to changes in RPE.

(2) Similarly, citrate synthase (CS) was found to be almost exclusively expressed in photoreceptor inner segments, with little expression in RPE (Figure 3D). Hence, the quantitative increase of citrate levels after FCCP treatment can only originate in photoreceptors.

(3) The comparison of the control (with RPE) against the no RPE group suggested an increase in (aerobic) glycolysis in the absence of RPE, evidenced notably by a retinal accumulation of lactate, BCAAs, and glutamate (Figure 3A). The very same metabolite pattern is seen for the FCCP treatment (Figure 1B) indicating a marked upregulation of glycolysis (Figure 6C). The latter observation suggests that photoreceptors, after disruption of OXPHOS switch to an exclusively glycolytic metabolism, which, however, rods cannot sustain (Figure 1C, D).

(4) Glucose consumption and lactate release is increased in the no RPE group vs. control (new Supplementary Figure 4). A similar increase in glucose consumption and lactate production is seen in the FCCP group suggesting that also the no RPE situation disrupts OXPHOS in photoreceptors.

Rev#1.10. The conclusions being reached are difficult to interpret secondary to the experimental procedures and the fact that the treatments are not cell-specific and RPE is included with the neuroretina as well. Likewise, stating FCCP is altering the Krebs cycle in the neuroretina is difficult to believe as there are no changes in the Krebs cycle when compared to the control, which also has RPE.

Response: We agree with the reviewer, that some of the conclusions may have been somewhat speculative. Accordingly, we have toned down our conclusions in several instances in the text, notably in abstract, introduction, and discussion.

When it comes to Krebs cycle intermediates a key limitation of our study is indeed the lack of carbon-tracing and metabolic flux analysis as noted by the reviewers, a limitation that we now highlight more strongly in the discussion of the revised manuscript (Lines 545-549). While it is highly probable that the flux of Krebs cycle intermediates is altered by FCCP, our steady-state data does not show significant changes in the metabolites citrate, fumarate, and succinate. However, our study does show a highly significant decrease in GTP levels, which as explained above, is a key indicator of Krebs cycle activity/inactivity. Moreover, while GTP levels were reduced also in the no RPE group, GTP was still significantly higher in the no RPE group compared to the FCCP treatment. Our interpretation of this finding is that there is Krebs-cycle/OXPHOS activity in the neuroretina, which is abolished by FCCP.

Rev#1.11. Supplemental Figure 4C and D states that GAC inhibition affected only photoreceptors, but GAC is expressed throughout the retina and so the inhibition is altering glutamine-glutamate homeostasis throughout the retina. Clearly, based on histology, one can see that the architecture of the retina, especially at the highest dose, is lost likely because all cells are being affected. So it is not photoreceptor-specific and even at low doses one can see that the inner retina is edematous. Moreover, with such a high amount of TUNEL staining in the ONL, are rods more affected than cones?

Response: In our hands the immunostaining for Glutaminase C (GAC) labelled predominantly cone inner segments, the OPL, and perhaps bipolar cells (Figure S1A). The deleterious effects mentioned by the reviewer are only seen at the highest concentration of the GAC inhibitor compound 968. This concentration (10 µM) is 100-fold higher than the dose that produces a significant loss of cones in the outer retina (0.1 µM). We therefore think that this data points to the extraordinary reliance of cones on glutamine and glutamate. As can be seen from the images (Figure S4C) illustrating the effects of 0.1 and 1 µM Compound 968 treatment, the ONL thickness is not significantly reduced by the GAC inhibitor. This strongly indicates that at these doses the rods are not affected by GAC inhibition.

Rev#1.12. The no RPE vs 1,9 DDF data may be interpreted as preventing glucose transport in the RPE increases BCAA catabolism by the RPE, which has been shown to utilize BCAA in culture systems. To this end, when the RPE is not present, the BCAA is increased as compared to the control with RPE.

Response: Our original interpretation of this data was that after GLUT1 inhibition and a correspondingly reduced retinal glucose uptake, the retina switched to an increasing use of anaplerotic substrates, including BCAAs. This is supported by the concomitant upregulation of the Cahill-cycle product alanine and the mini-Krebs-cycle product N-acetylaspartate (NAA). Yet, we agree with the reviewer that BCAAs could also be consumed by the RPE. We have now changed our conclusion at the end of the results chapter “Reduced retinal glucose uptake promotes anaplerotic metabolism“ to also highlight this possibility (Lines 261-262).

Rev#1.13. It is unclear why so much effort is comparing the no RPE group to the treatment groups and not comparing the control group to the different treatment groups.

Response: Previous studies – including the seminal studies of Otto Warburg from the early 1920s – had always used retina without RPE. This “no RPE” situation is therefore something of a reference for our entire study, which is why we dedicated more effort to its analysis. We have now inserted a corresponding remark into the manuscript (Lines 182-184).

Rev#1.14. The conclusions are significantly overstated especially with regards to rods versus cones as these are not cell-specific treatments. For example, the control vs 1,9 DDF vs FCCP clearly shows that there is mitochondrial dysfunction due to decreased NAD, increased AMP/ATP ratio, decreased Asp but increased Gln, and a compensatory increase in lactate production.

Response: We agree with the reviewer and have tried to phrase our statements in more measured fashion. Notably, we have toned down our statements in the title, abstract, results, discussion, and several of the subchapter headings.

Rev#1.15. While metabolic conclusions are drawn on serine/lactate ratio, this ratio is driven by the drastic changes in lactate and not so much serine in the treatment conditions as it was rather stable. Likewise, substrates beyond glucose have the potential to fuel the TCA cycle and make GTP via SUCLG1, such as fatty acids, other AAs, etc. Therefore, this ratio may not tell the entire story about anaplerotic metabolism. Furthermore, knowing that RPE utilize BCAAs to fuel their TCA cycle, the no RPE condition may simply have increased BCAAs due to lack of metabolism by the RPE, which drives the GTP/BCAA ratio. To state that the neuroretina was utilizing BCAAs for anaplerosis is not well supported based on the current data. Similarly, what is to say that the GTP/lactate ratio in the no RPE situation is not driven by the fact that the RPE is no longer present to act as acceptor of retinal lactate production or that more glucose is reaching the retina since the RPE is not present to accept and utilize that produced. Glucose uptake was not assessed to further address these issues.

Response: We agree with reviewer that metabolite ratios may not tell the full story underlying retinal metabolism however based on the robustness of using quantitative and highly reproducible NMR data, they are an important part of the metabolomics toolbox. The reviewer correctly observed that the changes in lactate levels are more dramatic than in serine. Still, also serine was significantly increased in the no RPE, 1,9-DDF, and Shikonin groups. Together with the lactate changes (same or opposite direction) the resulting serine/lactate ratios display marked alterations.

When it comes to the supply of other potential energy substrates mentioned by the reviewer, i.e. fatty acids or amino acids other than BCAAs, these are only supplied in minimal amounts in the defined, serum free R16 medium (Romijn, Biology of the Cell, 63, 263-268, 1988) and – if used to any important extent – would be rapidly depleted by the retina. Thus, for a culture period of 2 days in vitro between medium changes these energy sources are not available and thus cannot be used by the retina.

Our conclusion that the retina is using anaplerosis is based not only on the observations made in the no RPE group but also on, for instance, the metabolite ratios seen in the 1,9-DDF treatment group. In this group decreased glycolytic activity may correspond to increased serine synthesis and anaplerosis.

As far as glucose uptake is concerned, we have analysed the medium samples at P15 (equivalent to the retina tissue collection time point) and now present data that addresses this question more directly via the consumption of glucose from and release of lactate to the culture medium (New Supplementary Figure 4C, D). This new dataset provides another independent observation showing that:

(1) Glucose consumption/lactate release (i.e. aerobic glycolysis) is high in the no RPE situation but low in the control situation. In other words, retinal aerobic glycolysis is most likely stimulated by the absence of RPE.

(2) 1,9-DDF treatment decreases glucose consumption/lactate release as would be expected from a GLUT1 blocker. Since ATP and GTP production are high nonetheless, this indicates that other substrates (i.e. anaplerosis) were used for retinal energy production, in agreement with the analysis shown in Figure 6C.

(3) The FCCP treatment, which disrupts oxidative ATP-production, increases glucose consumption/lactate release in way similar to the no RPE situation. Yet, the no RPE retina can still generate sizeable amounts of GTP but not ATP. Together, this provides further evidence that neuroretinal OXPHOS is decreased in the absence of RPE.

Rev#1.16. The evidence for the mini-Krebs cycle is intriguing but weak considering it is based on certain enzymes being expressed in the photoreceptors, which had already been shown to be present in other publications, and a single ratio of metabolites that is increased in FCCP. One would expect this ratio to be increased under FCCP regardless. There is no stable isotope tracing with certain fuels to confirm the existence of the mini-Krebs cycle.

Response: We thank the reviewer for this suggestion. We agree that our evidence for the mini-Krebs-cycle (and the Cahill-cycle) may be to some extent circumstantial and additional technologies would help to obtain further supportive data. Still, here we would like to invite the reviewer to a thought experiment where he/she could try and interpret our data without considering the Cahill- or the mini-Krebs-cycle. At least we ourselves, when we engaged into such thought experiments, were unable to explain the data observed without these alternative energy-producing cycles. Most notably, we were unable to explain the strong accumulation of either alanine or N-acetyl-aspartate (NAA) when only considering glycolysis and (full) Krebs-cycle metabolism. Of course, this may still be considered “weak” evidence, and we expect that future studies including complementary technologies will either confirm or expand our interpretation of the existing data set.

The suggestion to perform stable isotope-labelled tracing with potential alternative fuels (e.g. glutamate, glutamine, pyruvate, etc.) is very attractive indeed. While such studies are likely to shed further light on the metabolic pathways proposed, this will entail very extensive experimental work, with multiple different conditions and concentrations and variety of analysis methods that is currently not feasible (e.g. a 1.7 mm NMR probe equipped with a 15N channel) as an extension of the present manuscript. Nevertheless, we will certainly consider this approach for future follow-up studies once such techniques are available and will screen for suited collaboration partners. A corresponding comment on such future possibilities has now been inserted into the discussion (Lines 545-549).

Rev#1.17. The discussion does not mention how this data contradicts a recent in vivo study looking at Glut1 knockout in the retina (Daniele et al. FASEB. 2022) or previous in vivo studies that suggest cones may be less sensitive to changes in glucose levels (Swarup et al. 2019). This is a key oversight.

Response: We thank the reviewer for pointing this out. We now included these studies in the revised discussion in a new subchapter on the expression of glucose transporters in the outer retina (Lines 454-472). For a critical review of the Daniele et al., 2022 study please also see our more detailed response to question Rev#3.13 below.

Rev#1.18. GAC is expressed in more than just cones so making cell-specific statements regarding fuel utilization is not well supported.

Response: Our immunostaining for GAC revealed a strong expression in cone inner segments (Figure S1A3). While this does not exclude (relatively minor) expression in other retinal cell types, cones are likely to be more reliant on GAC activity than other cell types. See also answer above.

Rev#1.19. Suggesting that rods utilize the mini-Krebs cycle based on AAT2 being seen in the inner segments without at least co-staining for RHO or PNA is weak evidence for such a cycle. AAT looks to be expressed in the inner segments of all photoreceptors.

Response: We have taken up this suggestion from the reviewer and now provide an additional co-staining for AAT1 and AAT2 with rhodopsin. Note that in response to a pertinent comment from Reviewer #3 we have changed the abbreviation for aspartate aminotransferase from “AAT” to the more commonly used “AST” throughout the manuscript.

New images showing a co-staining for AST1 and AST2 with rhodopsin now replace the former image set in Figure 7D. In brief, the new images show the expression of both AST1 and AST2 across the retina, with, notably an expression in the inner segments of photoreceptors but not in the outer segments, where rhodopsin is expressed.

**Reviewer #3 (Recommendations For The Authors):**
Rev#3.1. The staining for the glucose transporters GLUT1 and GLUT3 does not reflect what has previously been published by two different groups that were validated by cell-specific knockout mice. As mentioned by the author GLUT1 and GLUT3 have differences in transport kinetics, which would affect their metabolism. Therefore, the lack of GLUT1 in photoreceptors would suggest that photoreceptor metabolism is not faithfully replicated in this system. This difference from the previous literature should be discussed in the discussion.

Response: As the reviewer pointed out, the expression of GLUT1 in the retina is somewhat controversial, with much older literature showing expression on the RPE, while some more recent studies claim GLUT1 expression in photoreceptors. For a brief discussion of our GLUT1 immunostaining please see also our answer to question Rev#1.3 above.

Although the retinal expression of GLUT1 was besides the focus of our study, we feel we must address this point in more detail: In the brain the generally accepted setup for GLUT1 and GLUT3 expression is that low-affinity GLUT1 (Km = 6.9 mM) is expressed on glial cells, which contact blood vessels, while high-affinity GLUT3 (Km = 1.8 mM) is expressed on neurons (Burant & Bell, Biochemistry 31:10414-20, 1992; Koepsell, Pflügers Archiv 472, 1299–1343, 2020). This setup matches decreasing glucose concentration with increasing transporter affinity, for an efficient transport of glucose from blood vessels, to glial cells, to neurons. In the retina, the cells that contact the choroidal blood vessels are the tight-junction-coupled RPE cells. As shown by us and many others, RPE cells strongly express GLUT1 (cf. Figure 1A-3.). To warrant an efficient glucose transport from the RPE to photoreceptors, photoreceptors must express a glucose transporter with higher glucose affinity than GLUT1. We show that this is indeed the case with photoreceptors expressing GLUT3 (cf. Supplemental Figure 1-5.). While a part teleological explanation does not per se prove that our data is correct, at least our data is plausible. In contrast, the glucose transporter setup sometimes claimed in the literature is biochemically implausible, i.e. for the flow of metabolites (glucose) to go against a gradient of transporter affinities, and we are not aware of an example of such a setup occurring anywhere in nature.

However, at this point we cannot exclude low levels of GLUT1 expression on Müller glia cells or even photoreceptors. This expression could, for instance, be relevant in cases where cells were shuttling excess glucose – perhaps produced through gluconeogenesis – onwards to other retinal cells. Still, GLUT1 expression can only be minor when compared to RPE since a major expression would destroy the glucose affinity gradient (see above) required for efficient glucose shuttling into the energy hungry photoreceptors.

To address this request by the reviewer (and also reviewer #1) we now discuss the question of glucose transporter expression in the outer retina in a new subchapter of the discussion (Lines 454-472).

Rev#3.2. Photoreceptor metabolism and aerobic glycolysis are tied to photoreceptor function, as demonstrated by Dr. Barry Winkler. The authors should provide data or mention (if previously published) about photoreceptor OS growth and function in this system.

Response: The studies of Barry Winkler (e.g. Winkler, J Gen Physiol. 77, 667-692, 1981) confirmed the original work of Otto Warburg and expanded on the idea that the neuroretina was using aerobic glycolysis. Importantly, Winkler used a very similar experimental setup as Warburg has used, namely explanted rat retina without RPE. In light of our data where we compare metabolism of mouse retina with and without RPE – where retina cultured without RPE confirms the data of Warburg and Winkler – it appears most likely that the purported aerobic glycolysis occurs mostly in the absence of RPE but only to a lower extent in the native retina.

Photoreceptor outer segment outgrowth is somewhat slower in the organotypic retinal explant cultures compared to the in vivo situation (cf. Caffe et al., Curr Eye Res. 8:1083-1092, 1989 with LaVail, JCB 58:650-661, 1978; see also answer to reviewer #1). Importantly, organotypic retinal explant cultures and their photoreceptors are fully functional and remain so for extended periods in culture (Haq et al., Bioengineering 10:725, 2023; Tolone et al., IJMS 24:15277, 2023). This information has now been added to the manuscript discussion section, into the new subchapter “The retina as an experimental system for studies into neuronal energy metabolism” (Lines 367-395).

Rev#3.3. It is unclear from the description of the experiment in both the results and methods if 1,9DDF, Shikonin, and FCCP were added to both apical and basal media compartments or one or the other and should be specified. The details of what was on the apical compartment would be helpful, as the model is supposed to allow for only nutrients from the basal compartment (as indicated by the authors themselves). Is the apical compartment just exposed to air? How does this affect survival?

Response: In organotypic retinal explant cultures the RPE rests on the permeable culturing membrane such that the basal side is contact with the membrane and the medium below (far schematic drawing see Figure S1B), while the apical side is covered by a thin film of medium created by the surface tension of water (Caffe et al., Curr Eye Res. 1989; Belhadj et al., JoVE, 2020). This thin liquid film ensures sufficient oxygenation and is an important factor that allows the retinal explant to remain viable for several weeks in culture. If the retinal cultures were submerged by the medium, their viability – especially that of the photoreceptors – would drop dramatically and would typically be below 3-5 days. Therefore, in the retinal organotypic explant cultures used here, the nutrients and the drugs applied do indeed reach the outer retina from the basal side, i.e. similar as they would in vivo.

To address this question from the reviewer, corresponding clarifications have been inserted into the SI Materials and Methods section (Lines 64-66).

Rev#3.4. As the metabolomic data obtained was quantitative, several metabolites discussed should be analyzed in terms of ratios, for example, Glutathione and glutathione disulfide should be reported as a ratio. In addition as ATP, ADP, and AMP were measured, they can used to calculate the energy charge of the tissue.

Response: We thank the reviewer for these suggestions and have created corresponding graphs for GSH / GSSG ratio and energy charge. These new graphs have now been added to the SI datasets, to the new Supplementary Figure 4. To accommodate other requests from the Reviewers, this new Figure also contains additional new datasets on glucose and lactate concentrations (see further comments above and below). Please note that all later SI Figures have been renumbered accordingly.

In brief, the ratios for GSH/GSSG show no significant changes between control and the different experimental groups. Meanwhile, the adenylate energy charge of the retinal tissues show a significant decrease in the energy charge for the Shikonin group and the FCCP group. Note that in the new Supplementary Figure 4A, the dotted lines indicate the energy charge window typical for most healthy cells (0.7 – 0.95).

Rev#3.5. I think a missed opportunity when discussing the possible taurine/hypotaurine shuttle would be the impact on the osmosis of the subretinal space as taurine has been hypothesized as a major osmolyte.

Response: This is another interesting recommendation from the reviewer. To address this point, we have now introduced a corresponding paragraph and references in the discussion of the manuscript (Lines 503-504; 512-514).

Rev#3.6. In Figure 3, the distribution of these enzymes should also be studied under the no RPE condition as the culture treatment took several days for these metabolic changes to occur.

Response: The images shown in Figure 3D are from the in vivo retina. Since this may not have been very clear in the previous manuscript version, we have now added a corresponding explanation to the legend of Figure 3. As far as we can tell, the expression and localization of neuroretinal enzymes does not change in cultured retina, during the culture period (compare Figure 1A with Supplementary Figure S1C). However, when it comes to the metabolite taurine its production (localization) changes dramatically in the no RPE situation where taurine is essentially undetectable by immunostaining (not shown but see metabolite data in Figure 2A, Figure 3A).

Rev#3.7. In Figures 4 and 5, it is unclear why the experimental groups were not compared to the control and requires further explanation. Furthermore, the authors should justify the concentrations of drugs used as the cell death could have risen from toxicity to the drugs and not due to disruption of metabolism.

Response: The reviewer is right, the rationale for these comparisons may not have been laid out with sufficient clarity. In Figure 4 the no RPE and FCCP groups are compared because both groups showed similar metabolite changes towards the control situation. The no RPE to FCCP comparison thus focussed on the details of the – at first seemingly minor – differences between these two groups. This has now been clarified in the corresponding part of the results (Lines 218-219).

In Figure 5A, B we compare the no RPE and 1,9-DDF groups with each other, notably because the data obtained seemingly contradicted our initial expectation that these two groups should show similar metabolite patterns. Also here, we have now inserted an additional explanation for this choice of comparisons (Lines 252-253).

In Figure 5C, D we compare the Shikonin and FCCP groups with each other. The idea behind this comparison was that in the 1st group glycolysis was blocked while in the 2nd group OXPHOS was inhibited, or in other words here were compared what happened when the two opposing ends of energy metabolism were manipulated in opposite directions. This reasoning is now given in the results section (Lines 265-268).

As far as the choice of drugs and concentrations is concerned, we used only compounds that have been extremely well validated through up to five decades of scientific research (see initial response to Reviewer #2 above). We therefore are confident that at the concentrations employed the results obtained stem from drug effects on metabolism and not from generic, off-target toxicity. Then again, as we show, prolonged (i.e. 4 days) block of energy metabolism pathways does cause cell death.

Rev#3.8. In line 203, the authors discuss GTP as being primarily a mitochondrial metabolite, however, photoreceptors would require a localized source of GTP synthesis in the outer segments as part of phototransduction, and therefore GTP in photoreceptors cannot be a mitochondrial-specific reaction in photoreceptors. Furthermore, the authors mentioned NDK as being a possible source of GTP, but they do not show NDK localization despite it being reported in the literature to be localized in the OS.

Response: The question as to the source of GTP in photoreceptor outer segments is indeed highly relevant. For GTP production in mitochondria see the answer to the next question below (Rev#3.9). An early study showed nucleoside-diphosphate kinases (NDK) to be expressed on the rod outer segments of bovine retina (Abdulaev et al., Biochemistry 37:13958-13967, 1998). More recently NDK-A was shown to be strongly expressed in photoreceptor inner segments (Rueda et al., Molecular Vision 22:847-885, 2016). We now refer to both studies in the results section of the manuscript (Line 227-228).

Rev#3.9. In the "Impact on glycolytic activity, serine synthesis pathway, and anaplerotic metabolism" section, the authors claim in the no RPE group glycolytic activity was higher due to a depressed GTP-to-lactate ratio. However, this reviewer is under the impression that GTP production in photoreceptors is not mitochondrial specific, so this ratio doesn't make sense (I could be mistaken, however). A better ratio would have been pyruvate/lactate or glucose/lactate when discussing increased glucose consumption.

Response: We appreciate the reviewers’ comment, yet we do indeed believe we can show that GTP-production in our experimental context is mainly mitochondrial. As explained in the manuscript results section (“Photoreceptors use the Krebs-cycle to produce GTP”), there are essentially only two possibilities for a photoreceptor to produce sizeable amounts of GTP. In the mitochondria via SUCLG1 – i.e. an enzyme highly expressed in photoreceptor inner segments (Figure 5D) – and the cytoplasm via NDK from excess ATP. The claim about the depressed GTP-to-lactate ratio in the no RPE situation takes this into account. Importantly, since in the no RPE situation ATP-levels are significantly lower than GTP, here GTP can only be produced via SUCLG1 and OXPHOS. Moreover, this contrasts with the FCCP group where mitochondrial OXPHOS is disrupted and both ATP and GTP are depleted.

As far as ratios with pyruvate and glucose are concerned, we agree that these could potentially be very interesting to analyse. Unfortunately, in our retinal tissue 1H-NMR spectroscopy- based metabolomics analysis the levels of both pyruvate and glucose were below the detection limits which likely reflects their rapid metabolic turnover (cf. table S1). While this might be attributable to the marked consumption of these metabolites within the tissue, it does not allow for us to calculate the suggested ratios to lactate. Then again, in the supernatant medium which was collected at the same time point as the retina tissue, we can readily detect glucose and lactate levels, for this data please see the new Supplementary Figure 4.

Rev#3.10. Aspartate aminotransferase should be abbreviated as AST, as it is more commonly noted.

Response: In response to this comment from the reviewer, we have changed the abbreviation for aspartate aminotransferase from AAT to AST throughout the manuscript.

Rev#3.11. In the discussion the assumptions of the ex vivo culture systems should be clearly stated. One that was not mentioned, but affects the implications of the data, is that the retinas used in this study are from the developing mouse eye. Another important assumption that was made in this paper was that the changes in retinal metabolism were due to photoreceptors even though the whole neural retina was included.

Response: The reviewer is correct; we have added these two points to the discussion section of the manuscript. Notably, we now included a new subchapter “The retina as an experimental system for studies into neuronal energy metabolism” (Lines 367-395) to present different in vitro and in vivo test systems.

Rev#3.12. Starting at line 347: As the authors know, the RPE has been shown to be highly reliant on mitochondrial function, and disruption of RPE mitochondrial metabolism leads to photoreceptor degeneration (numerous papers have shown this). Furthermore, the lower levels of lactate detected in their explants when RPE was present suggests that lactate is actively transported out of the neural retina by the RPE.

Response: The reviewer is right about lactate being exported from the retina to the blood stream in vivo, or, in our in vitro study, to the culture medium. In the new dataset showing glucose and lactate concentrations in the culture medium (new Supplementary Figure 4C, D), we show that without RPE (no RPE group) and the retina releases more significantly lactate into the medium than control retina with RPE. At the same time the no RPE retina consumes more glucose than control retina.

Rev#3.13. Line 360: Again, in mouse photoreceptors (by bulk RNAseq and scRNAseq), there is no GLUT3 expression (encoded by slc2a3). It was also recently shown by Dr. Nancy Philp's lab that rod photoreceptors express GLUT1, encoded by slc2a1 (PMCID: PMC9438481). The differences reported in this study and previous studies should be discussed.

Response: Although this comment may not make us very popular, we are somewhat sceptical of RNAseq data (especially single cell RNAseq) since the underlying methodology – at the current level of technological development – is notoriously unreliable when it comes to the assessment of low abundance transcripts and suffers from apoor batch reproducibility, compared to NMR based metabolomics. Due to methodological constraints RNAseq have a propensity to display erroneously high or low expression. Moreover, and perhaps even more important, dissociated cells in scRNAseq studies undergo rapid gene expression changes that can significantly falsify the image obtained (Rajala et al., PNAS Nexus 2:1-12, 2023). Finally, it cannot be emphasized enough that mRNA expression profiles DO NOT equate protein expression and there are numerous examples for divergent expression profiles when mRNA and protein is compared.

The Daniele et al. study (FASEB Journal 36:e22428, 2022; PMCID: PMC9438481) used in situ hybridization to study the mRNA expression of GLUT1 (slc2a1) and GLUT3 (slc2a3). In line with our comment just above, the Daniele et al. study may provide for an example of divergence between mRNA and protein expression, since it seemingly showed only minor expression of GLUT1/slc2a1 in the RPE, i.e. precisely in the one cell type that is well-known for its very strong GLUT1 protein expression.

Furthermore, Daniele et al. used a conditional GLUT1 knock-out in photoreceptors induced by repeated Tamoxifen injections. The photoreceptor GLUT1 knock-out led to a relatively mild phenotype with only about 45% of the outer nuclear layer lost over a 4-months time-course. This is in stark contrast with the FCCP or the 1,9-DDF treatment, which would ablate nearly all rod photoreceptors in under one or two weeks, respectively.

As a side note, Tamoxifen is an oestrogen receptor antagonist (with partial agonistic behaviour) with a long history of causing retinal and photoreceptor damage. Notably, oestrogen receptor signalling is important for maintaining photoreceptor viability (Nixon & Simpkins, IOVS 53:4739-47, 2012; Xiong et al., Neuroscience 452:280-294, 2021). Therefore, the relatively minor effects of the conditional GLUT1 KO in photoreceptors found in Daniele et al. may have been confounded by direct tamoxifen photoreceptor toxicity. On a wider level, this possible confounding factor related to the use of Tamoxifen points to general problems associated with certain forms of genetic manipulations.

We now mention the controversy around the expression of glucose transporters in the retina, including the Daniele et al. study in a new subchapter of the discussion on "Expression of glucose transporters in the outer retina” (Lines 454-472).

Rev#3.14. Lines 370-372: FCCP caused a strong cell death phenotype in rods, however under stress rods upregulate the secretion of RdCVF, which leads to cone photoreceptor survival by the upregulation of aerobic glycolysis in cones. The data should be re-interpreted in the context of this previous literature.

Response: We thank the reviewer for this comment; however, we could not find a reference that would state that “…under stress rods upregulate the secretion of RdCVF”. What we did find was a reference stating that similar factors such as thioredoxins (TRX80) are secreted from blood monocytes under stress (Sahaf & Rosén, Antioxid Redox Signal 2:717-26, 2000). However, we consider these cells to be too dissimilar to rod photoreceptors to warrant a corresponding comment. Moreover, the research group who discovered RdCVF originally showed that rod-secreted RdCVF cannot prevent cone degeneration if the corresponding Nxnl1 gene is knocked-out in cones, arguing for a cell-autonomous mechanism of RdCVF -dependent cone protection (Mei et al., Antioxid Redox Signal. 24:909-23, 2016).

Since it is very possible that we may have missed the correct reference(s), we would welcome further guidance by the reviewer.

Rev#3.15. Line 374: 1,9-DDF caused a 90% loss of cones, however, previous studies by Dr. Nancy Philp have shown glucose deprivation in the outer retina affects primarily rod photoreceptors. The differences should be discussed.

Response: We thank the reviewer for directing us to these studies. As mentioned above (Rev#3.13.) the Daniele et al. 2022 study yielded only relatively mild effects for a rod-specific conditional GLUT1 KO on photoreceptor viability. Similarly, in an earlier study (Swarup et al., Am J Physiol Cell Physiol. 316: C121–C133, 2019) the Philp group found that also a GLUT1 KO in the RPE caused only a minor phenotype in the photoreceptor layer. We would argue that if glucose, and by extension aerobic glycolysis, were indeed of major importance for (rod) photoreceptor survival, the degenerative effect of these genetic GLUT1 ablations should have been devastating and should have destroyed most of the outer retina in a matter of days. The fact that this was not seen in both studies is another piece of independent evidence that rod photoreceptors do not rely to any major extent on glycolytic metabolism.

The two studies from the Philp lab (Swarup et al., 2019; Daniele et al., 2022) are now cited in the discussion (Lines 417-419 and 458-460).

Rev#3.16. Line 375: Yes Dr. Claudio Punzo and Dr. Leveillard Thierry along with other groups have shown glycolysis is required to maintain cone survival when under stress, however, the authors should emphasize that it is under stress that this is observed.

Response: In response to this comment we have now specifically extended our corresponding remark in the discussion of the manuscript (Lines 446-447).

Rev#3.17. The section "Cone photoreceptors use the Cahill-cycle". The presence of ALT in photoreceptors was surprising and suggests alternatives to the Cori reaction. However, previous measurements of glucose and lactate from localized in vivo cannulation of animal eyes suggest the majority of glucose taken up by the retina is released back to the blood as lactate. Again, this section should discuss this idea in terms of the previous literature.

Response: Here, we believe the reviewer is referring to studies performed in the late 1990s where, in anaesthetized cats, the lactate concentration in blood samples obtained from choroidal vein cannulation was compared against that in blood samples obtained from femoral arteries (Wang et al., IOVS 38:48-55, 1997). We note that a more relevant in vivo measurement of retinal glucose consumption and lactate production would likely require the simultaneous cannulation of the central retinal artery (CRA) and the central retinal vein (CRV). This would need to be combined with repeated (online) blood sampling, drug applications, and subsequent metabolomic analysis. We are not aware of any in vivo studies where such procedures have been successfully performed and further miniaturization and increased sensitivity of metabolomic analytic equipment will likely be required before such an undertaking may become feasible. Even so, such studies may not be feasible in small rodents (mice, rats) and may instead require larger animal species (e.g. dog, monkey) to overcome limitations in eye and blood sample size.

We have now extended the discussion of our manuscript with a new subchapter on “The retina as an experimental system for studies into neuronal energy metabolism”. Within this new subchapter we now present two different in vivo experimental approaches that addressed retinal energy metabolism (Lines 376-384). Moreover, we now present new data on retinal lactate release to the culture medium, showing, for instance, a strong increase in lactate release in the no RPE condition compared to control (new Supplementary Figure 4).

Rev#3.18. Lines 431-433: The study cited suggested that the mitochondrial AST was detected in other cells, in agreement with the data shown. However, the authors' statements in this section are misleading as they do not take into consideration the contribution of AST from other cell types.

Response: The reviewer is right, we found both AST1 and AST2 to be expressed not only in photoreceptor inner segments but also in the inner retina, especially in the inner plexiform layer (new Figure 6D). Since this might indicate mini-Krebs-cycle activity also in retinal synapses, we have added a corresponding comment to the discussion (Lines 540-543).

Grammatical and wording fixes:Rev#3.19. Line 98 - "the recycling of the photopigment, retinal."

Response: We have inserted a comma after “photopigment”.

Rev#3.20. Results section and Figure 1 start without providing context for the model system where staining is being done.

Response: We have added this information to the beginning of the results section (Lines 105-106).

Rev#3.21. Supplementary Figure 2 is not mentioned in the main text - there is no context for this figure.

Response: Supplementary Figure 2 was originally referenced in the legend to Figure 2. We now mention supplementary figure 2 (now renumbered to supplementary figure S3) also in the main text, in the results section under “Experimental retinal interventions produce characteristic metabolomic patterns” (Line 148).

Rev#3.22. Volcano plot in Supplementary Figures 3, 5, 6, 7, and 8 don't indicate what Log2(FC) is in reference to.

Response: The log2 fold change (FC) is calculated as follows: log2 (fold change) = log2 (mean metabolite concentration in condition A) - log2 (mean metabolite concentration in condition B) where condition A and condition B are two different experimental groups being compared. This is now explained in the SI Materials and Methods (Lines 145-147) and indicated in abbreviated form in the figure legends. Please note that supplemental figures have now been renumbered due to the insertion of an additional, new Figure.

Rev#3.23. Line 331 - –a“d allowed to analyze the..." ”s incorrect phrasing.

Response: This phrasing was changed.

Rev#3.24. Line 343 "c“cled" ”

Response: This phrasing was changed.

Rev#3.25. Line 446 is misworded.

Response: This phrasing was changed.

Technical questions:Rev#3.26. At what point after explant was the IHC done in Supplemental Figure 1? If early, but experiments are done later, there's’a chance things are more disorganized at the end of the experiment.

Response: Staining and metabolomics analysis were both done at the end of each experiment, at the same time, at P15. This is now mentioned in the SI materials and methods section (Lines 67, 108-110).

Rev#3.27. FCCP affects plasma membrane permeability, which is particularly critical in neurons that undergo repolarization and depolarization - –ow do we know FCCP on cell death via metabolism? See: https://www.sciencedirect.com/science/article/pii/S2212877813001233

Response: The reviewer is correct, a significant permeabilization of cell membranes in general would likely cause extensive neuronal cell death, unrelated to a disruption of OXPHOS. However, the FCCP concentration used here (5 µM) is at the lower end of what was used in the mentioned Kenwood et al. study (Mol Metab. 3:114-123, 2014) and the effect on cell membrane permeability in tissue culture is likely to be rather small, as opposed to what was seen by Kenwood et al. in cultures of individual cells. This view is supported by the fact that in our FCCP treatments, we did not observe any significant increases of cell death in any retinal cell type (including RPE) other than in rod photoreceptors. Together with the fact that only photoreceptors strongly express Krebs-cycle/OXPHOS related enzymes, this strongly suggests that the FCCP effects seen by us were due to disruption of OXPHOS.

Rev#3.28. Numerous metabolite comparisons are being made throughout the manuscript – what type of multiple hypothesis testing corrections are utilized? Only certain figures mention multiple hypothesis testing (e.g. Figure 6).

Response: In general, in this manuscript we used two different statistical methods: (1) For two-group comparisons, we used an unpaired, two-tailed t-test, which reports a p-value with 95% confidence interval without additional multiple hypothesis testing (e.g. in Figure 2, Suppl. Figures 4, 6, 7, 8). (2) For multiple group comparisons we used a one-way ANOVA analysis with Tukey’s multiple comparisons post-hoc test (except suppl. Figure 9 where Fisher´s LSD post-hoc test was used). The information on which statistical test was used for what dataset is now given in the figure legends and in the SI Material and Methods section.

Rev#3.29. For Figure 3, how do we know that the removal of RPE is causing the metabolite changes due to RPE-PR coupling? How do you rule out the fact that it isn’t just: I – a thicker physical barrier between media and the neural retina that is causing the changes, or II – removal of RPE from PR causes OS shearing and a stress response that alters metabolism?

Response: We believe these concerns can be ruled out: The RPE cells are linked by tight junctions and are not “just a thicker barrier” but a barrier that is almost impermeable for most metabolites unless they are carried by specific transporters. Outer segment shearing via RPE removal would indeed be a concern if we had used adult retina. However, we explanted that retina at P9 when it does not possess any sizeable outer segments yet. As a matter of fact, photoreceptors grow out outer segments only after P9.

Rev#3.30. While 1,9-dideoxyforskolin blocks GLUT1, it is known to have other effects, including on potassium channels. How do we know the effects of 1,9-dideoxyforskolin are specific to GLUT1? Utilizing a GLUT1 KO and showing no additional effects when adding 1,9-dideoxyforskolin would be helpful as a control.

Response: This is a good suggestion from the reviewer. We note that this is technically not easy to achieve as it would require an RPE-specific knock-out that should be inducible at a given experimental time-point, in a quantitative manner. The study by Swarup et al. (see above Rev#3.13.) used an RPE specific knock-out that was, however, not inducible. Moreover, if the corresponding inducible knock-out animals could be generated, then the stochastic nature of the inducing treatment would probably affect only a limited number of cells within a given cell population. In our experimental context, a less than quantitative knock-out would significantly complicate interpretation of results, even to the point that no additional insight might be gained.

Rev#3.31. The analysis in Figure 6, even with attempts to control drug treatments, is highly speculative. One really needs animals with predominately cones vs. predominately rods to do this analysis (e.g. with NRL mice).

Response: The reviewer is right, the analysis shown in Figure 6 was an explorative approach to try and deduce features of rod and cone metabolism. This is now mentioned in the results section (Lines 282-284). Since the experiments were not initially intended to address such questions, by necessity the interpretations remain speculative. The comparison of mouse mutants in which there are either no cones (e.g. cpfl1 mouse) or no rods (e.g. NRL knock-out mouse) may allow to disentangle the metabolic contributions of rods and cones. We appreciate the suggestion from the reviewer and have now inserted a relating suggestion for future studies into the discussion section (Lines 450-452).

Rev#3.32. Overall, much of the paper suggests intriguing pathways, but without C13 tracing or relevant genetic knock-outs, the pathways would have to be speculative rather than definitive.

Response: We agree with the reviewer that further research, including 13C and 15N-tracing studies, will be necessary to evaluate which pathway(s) are used by what retinal cell type under what condition. Still, the high robustness and quantitative nature of the NMR metabolomics data allows us to draw pathway conclusions based on metabolites that are unique to specific pathways/cell types or using ratios. We now relate to the advantages of such carbon-tracing studies in the discussion of the manuscript (Lines 545-549).

Stylistic suggestions:Rev#3.33. This is a very dense paper to read. It would be helpful for each figure to have a summary diagram of the relevant metabolite changes and how they fit together. Further, for those not metabolism-inclined, defining the mini-Kreb’s, Cahill, and Cori cycles and their brief implications at some point early in the manuscript would be helpful.

Response: We have been thinking a lot about how we could add in the suggested summary diagrams into each figure. Unfortunately, whatever idea we contemplated would have significantly increased the complexity of the figures, while the actual benefit in terms of improved understandability was unclear.

However, we did include the suggestion from the reviewer to present the terms Cori, Cahill-, and mini-Krebs-cycle already in the introduction and we hope that this has improved the understandability of the manuscript overall (Lines 79-92).

Rev#3.34. More discussion about the step-by-step ways that the mini-Kreb’s reaction “uncouples” glycolysis from the Kreb’s cycle would be helpful. What do you mean by “uncouple” in this context?

Response: We thank the reviewer for this suggestion. Uncoupling in this context means that glycolysis and Krebs cycle are not metabolically coupled to each other via pyruvate. Instead both pathways can run independently from each other and in parallel, as long as the Krebs-cycle uses glutamate, BCAAs or other amino acids as fuels. We now also address this point already in the introduction of the manuscript (Lines 87-90).

Conceptual questions:Rev#3.35. As the proposal that PR undergo heavy amounts of OXPHOS is controversial, it would be helpful for the authors to review the literature on lactate production by the retina and what studies have shown previously about retina use of lactate, specifically lactate making its way into TCA cycle intermediates, suggesting OXPHOS, in PRs.

Response: In response to this question we have added several new references to the introduction and discussion of the manuscript. The question of lactate production (aerobic glycolysis) vs. the use of OXPHOS is now discussed in Lines 77-81, Lines 367-384.

Rev#3.36. Why would cones die more in the no RPE condition? The authors suggest this has something to do with GLUT1 expression on RPE and the transport of glucose to cones. Even if we accept that cones are highly glycolytic, loss of RPE should expose the neural retina to even more glucose in your experimental set-up.

Response: This is a very interesting question from the reviewer. Indeed, loss of the RPE and blood-retinal barrier function should increase photoreceptor access to glucose, even more so if they are expressing high affinity GLUT3. In the discussion (Lines 420-424), we speculate that this may trigger the Crabtree effect, shutting down OXPHOS and causing the cells to exclusively rely on glycolysis. This, however, will likely not yield sufficient ATP to maintain their viability, so that they “starve” to death even in the presence of ample glucose. Since cones require at least twice as much ATP as rods, they may be more sensitive to a Crabtree-dependent shut-down of OXPHOS. However, if this speculation was correct then the question remains why the FCCP treatment, which abolishes OXPHOS more directly, does not cause cone death. Here, we again can only speculate that high glucose may have additional toxic effects on cones that are independent of OXPHOS. We now try to present this reasoning in the discussion (Lines 426-429).